# CRISPR screen for protein inclusion formation uncovers a role for SRRD in the regulation of intermediate filament dynamics and aggresome assembly

Katelyn M. Sweeney[1,2], Sapanna Chantarawong[1,2], Edward M. Barbieri[3], Greg Cajka[1,2], Matthew Liu[1,2], Lynn Spruce[4], Hossein Fazelinia[4,5], Bede Portz[3], Katie Copley[3], Tomer Lapidot[1,2], Lauren Duhamel[1,2], Phoebe Greenwald[1,2], Naseeb Saida[6], Reut Shalgi[6], James Shorter[3], Ophir Shalem[1,2]*

1 Center for Cellular and Molecular Therapeutics, Children's Hospital of Philadelphia, Philadelphia, Pennsylvania, United States of America, 2 Department of Genetics, Perelman School of Medicine, University of Pennsylvania, Philadelphia, Pennsylvania, United States of America, 3 Department of Biochemistry and Biophysics, Perelman School of Medicine, University of Pennsylvania, Philadelphia, Pennsylvania, United States of America, 4 Proteomics Core Facility, Children's Hospital of Philadelphia, Philadelphia, Pennsylvania, United States of America, 5 Department of Biomedical and Health Informatics, Children's Hospital of Philadelphia, Philadelphia, Pennsylvania, United States of America, 6 Department of Biochemistry, Rappaport Faculty of Medicine, Technion-Israel Institute of Technology, Haifa, Israel

* shalemo@upenn.edu

**Data Availability Statement:** All relevant data are within the paper and its Supporting information files.

## Abstract

The presence of large protein inclusions is a hallmark of neurodegeneration, and yet the precise molecular factors that contribute to their formation remain poorly understood. Screens using aggregation-prone proteins have commonly relied on downstream toxicity as a readout rather than the direct formation of aggregates. Here, we combined a genome-wide CRISPR knockout screen with Pulse Shape Analysis, a FACS-based method for inclusion detection, to identify direct modifiers of TDP-43 aggregation in human cells. Our screen revealed both canonical and novel proteostasis genes, and unearthed SRRD, a poorly characterized protein, as a top regulator of protein inclusion formation. APEX biotin labeling reveals that SRRD resides in proximity to proteins that are involved in the formation and breakage of disulfide bonds and to intermediate filaments, suggesting a role in regulation of the spatial dynamics of the intermediate filament network. Indeed, loss of SRRD results in aberrant intermediate filament fibrils and the impaired formation of aggresomes, including blunted vimentin cage structure, during proteotoxic stress. Interestingly, SRRD also localizes to aggresomes and unfolded proteins, and rescues proteotoxicity in yeast whereby its N-terminal low complexity domain is sufficient to induce this affect. Altogether this suggests an unanticipated and broad role for SRRD in cytoskeletal organization and cellular proteostasis.

## Author summary

The presence of large protein inclusions is a hallmark of many neurodegenerative diseases, yet the precise mechanisms by which cells compartmentalize unfolded proteins is

**Funding:** This work was supported by the following grants: NIH/NIGMS DP2GM137416 (OS, GC and ML, TL), PA DoH SAP#4100083086 (OS and SC, TL), NINDS/NIH R03NS111447-01 (OS), iAward Sanofi (OS), NINDS - F31NS116999 to KMS, The Packard Center (JS), TargetALS (JS), The Association for Frontotemporal Degeneration (JS), the Amyotrophic Lateral Sclerosis Association (JS), Office of the Assistant Secretary of Defense for Health Affairs through the Amyotrophic Lateral Sclerosis Research Program W81XWH-20-1-0242 and W81XWH-17-1-0237 (JS) G. Harold and Leila Y. Mathers Foundation (JS), NIH R01GM099836 (JS), NIH R21AG065854 (JS), EMB from (Milton Safenowitz Post-Doctoral Fellowship from the ALSA, NIH; BP from (AHA and the BrightFocus Foundation). The funders had no role in study design, data collection and analysis, decision to publish, or preparation of the manuscript.

**Competing interests:** I have read the journal's policy and the authors of this manuscript have the following competing interests: OS and KMS have filed a patent application for the use of SRRD fragments through the Children's Hospital of Philadelphia. J.S. is a consultant for Dewpoint Therapeutics, ADRx, and Neumora. J.S. is a shareholder and advisor for Confluence Therapeutics. The authors declare no other conflicts of interest relevant to this publication.

still not completely understood. Here we used a novel screening approach that enables FACS-based inclusion detection, to identify direct modifiers of TDP-43 aggregation in human cells. Our screen revealed both canonical and novel proteostasis genes, and unearthed SRRD, a poorly characterized protein, as a top regulator of protein inclusion formation. In follow up experiments, using both proximity labeling and imaging, we show that SRRD is involved in the regulation of intermediate filaments (IF) spatial organization. Loss of SRRD results in fragmented IF network and the reduced formation of aggresomes, which are transient compartments to which cells transfer unfolded proteins under cellular stress. Interestingly, without SRRD, aggresomes completely lack vimentin cages, which are typical structures that encapsulate aggresomes and have been associated with the recruitment of protein degradation and chaperones machinery to these structures. Altogether, we present a novel screening approach for the identification of gene associated with cellular proteostasis. We reveal several novel protein factors, including SRRD, which our data suggests has a broad and previously overlooked role cytoskeletal organization and protein quality control.

## Introduction

Cellular proteostasis refers to an array of cellular mechanisms that maintain the proteome in a folded and functioning state [1]. Cells harbor specialized molecular mechanisms to deal with the presence of misfolded proteins including regulation of protein translation, compartmentalization, folding, and degradation [2]. During unfolded protein stress, a number of quality control pathways are activated, such as chaperones and degradation machinery to alleviate protein misfolding and overabundance [3–5]. Additionally, synthesis of new proteins is paused, and the translation machinery is sequestered into transient cellular structures like stress granules. If chaperones or degradation machinery are overwhelmed, misfolded proteins are often sequestered into subcellular compartments to i) prevent their deleterious interactions with other cellular components, ii) enhance clearance at local sites with enriched chaperones, proteasomes, and autophagy machinery, or iii) terminally sequester insoluble proteins to promote asymmetric inheritance after cell division [4,5]. One example of such a subcellular compartment is the mammalian aggresome, a perinuclear body consisting of unfolded proteins that are actively recruited to the centrosome via HDAC6-coupled dynein-mediated trafficking [6–10]. Aggresomes are enriched in ubiquitinated proteins and chaperones [7,11], as well as in degradation machinery such as proteasomes and the autophagy adapter SQSTM1 (Johnston and Samant 2021), and they are encircled by cage-like structures formed from the intermediate filament protein Vimentin (VIM) [7,8,12].

The loss of proteostasis is a hallmark of many human diseases such as cancer, and neurodegenerative diseases and is thought to be directly related to the aging process [3–5]. In the case of neurodegenerative diseases, proteins that are prone to misfolding or contain intrinsically disordered regions misfold, mislocalize, and are unable to be cleared by degradation machinery. These proteinopathies are often associated with cellular toxicity, and thus to date many groups have employed genetic screens in yeast, *Drosophila*, and mammalian cells to find regulators of proteotoxicity [13–17]. These approaches will frequently capture downstream mechanisms of proteotoxicity that connect aggregation to cell death rather than providing mechanistic insight into aggregate formation.

In this study we used a unique approach to screen for novel regulators of protein inclusion formation directly, rather than the downstream protein toxicity. As a model for perturbed

proteostasis, we used TDP-43, an essential RNA-binding protein (RBP), which can be found in cytoplasmic aggregates that are a pathological hallmark of amyotrophic lateral sclerosis (ALS) and frontotemporal dementia (FTD) [18–21]. Mutations in TDP-43 are rare, explaining less than ~5% of ALS and FTD cases, and yet ~97% of ALS patients and ~50% of FTD patients present with TDP-43 pathology [22]. TDP-43 pathology can also be observed in cases of Alzheimer's disease and Parkinson's disease [21,23–25]. These findings suggest that aggregation of TDP-43 is heavily affected by many other proteins in the cell, making it a candidate to screen for modifiers of protein aggregation.

We used a fluorescent TDP-43 aggregation reporter that, when coupled to Pulse Shape Analysis (PulSA), an approach that uses flow cytometry to distinguish between a diffuse and punctate fluorescent signal in cells [26,27], enabled us to quantify TDP-43 aggregation at the single cell level. We leveraged this reporter with a genome-wide CRISPR-Cas9 KO screen which revealed both canonical proteostasis machinery and novel modifiers of TDP-43 aggregation. Our screen identified SRRD, a protein of unknown function, as a top positive regulator of TDP-43 inclusion formation. We further describe a role for SRRD as a regulator of intermediate filaments (IF) dynamics and aggresome formation. Using APEX proximity labeling we found that SRRD resides in close proximity to proteins IFs including multiple keratin proteins and vimentin, which has a well established role in aggresome formation. Interestingly, two molecular functions were highly enriched in the set of SRRD interactors: Protein disulfide-isomerase (PDI) and calcium binding. A calcium dependent formation and breakage of disulfide bonds has been shown to be essential for the regulation of IF structures [28], suggesting that SRRD may act as a regulator of IF spatial dynamics. Indeed, loss of SRRD results impaired vimentin organization in cells, lower protein expression of several cytoskeletal proteins, and impaired aggresome assembly characterized by lower aggresome formation and an almost complete lack of vimentin cages, under stress conditions. We also examined the localization of SRRD under proteotoxic stress and found that it localizes to both aggresomes and unfolded proteins, mediated by an N-terminal unstructured region suggesting it may also have a direct effect on unfolded proteins. Altogether, our work suggests a previously unappreciated role for SRRD in the regulation of cellular organization and proteostasis.

## Results

### PulSA pooled CRISPR genome-wide screen reveals known and novel mediators of cellular proteostasis

To facilitate FACS based CRISPR screening of protein inclusion formation directly, we first tested if Pulse-Shape Analysis (PulSA) [26], a method that distinguishes between diffuse and punctate fluorescent signal using fluorescence-width (duration) and fluorescence-height (intensity) (Fig 1A), can be used to detect TDP-43 cytoplasmic inclusions. We tested exogenous expression of several TDP-43 constructs, including wild type TDP-43 and TDP-43 with a deleted NLS sequence (TDP-43$_{\Delta NLS}$) fluorescently tagged on either the N- or C- terminus. While exogenous expression of wild type TDP-43 did not result in cytosolic inclusion formation (Figure A in S1 Text), deletion of the NLS resulted in cytoplasmic localization with the N-terminal tagged protein also yielding a subset of cells with perinuclear, punctate TDP-43 inclusions as previously described [18]. This was observed using both microscopy and FACS based PulSA (Figs 1B and Figure A in S1 Text). The C-terminal tagged TDP-43 also localized to the cytoplasm but did not form aggregates, likely due to the tag interfering with the C-terminal prion-like domain of TDP-43, which drives aggregation [29]. We thus used this construct as a control to show the PulSA pattern of cytoplasmically localized TDP-43$_{\Delta NLS}$ that does not form aggregates. We previously demonstrated that sorting based on PulSA was accurate enough to

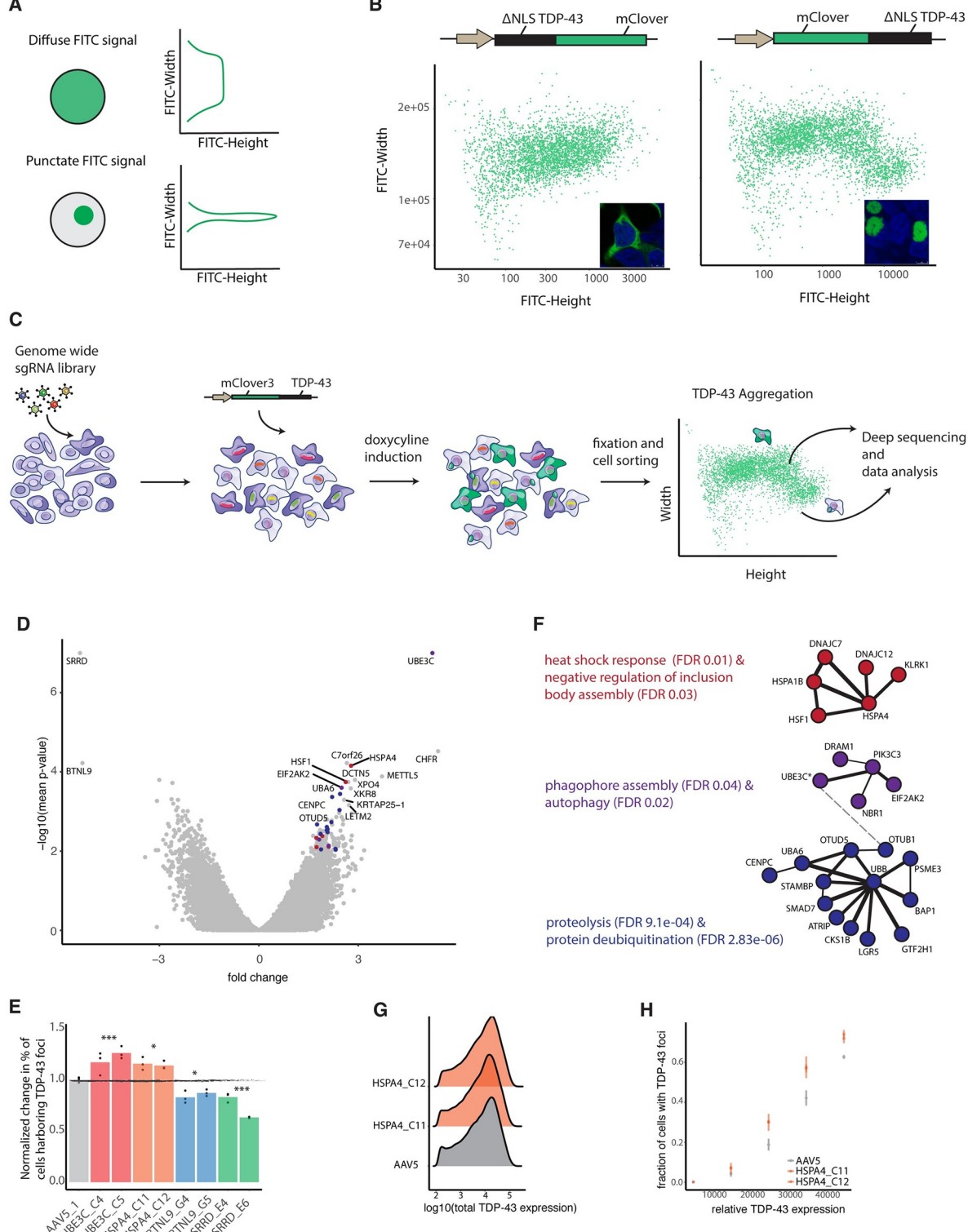

**Fig 1. Genome-wide CRISPR-Cas9 KO screen using Pulse-Shape Analysis uncovers expected and novel modifiers of cellular proteostasis.** A) Schematic of theory underlying PulSA. B) Schematic of TDP-43 $_{\Delta NLS}$ expression vectors tagged C- or N- terminally with mClover3, PulSA plots generated after transfection of corresponding vectors in 293Ts, and inset of cells showing TDP-43 localization. C) Schematic of genome-wide CRISPR-Cas9 KO screening method. D) Volcano plot of CRISPR screen results where each dot is the average of 4 sgRNAs targeting each gene, displaying the phenotype (x-axis) vs the significance (y-axis). Colored dots correspond to STRING cluster in

1F. E) Arrayed screen validation of top modifiers of TDP-43 $_{\Delta NLS}$ aggregation. Each sgRNA KO is normalized to AAV5 non-targeting control and plotted to show the normalized change in the % of cells harboring TDP-43 $_{\Delta NLS}$ aggregates. n = 3 replicates, one way anova with Tukey HSD test comparing AAV control to all 6 sgRNAs targeting each gene. Adjusted p -values: AAV-UBE3C 0.00038; AAV-HSPA4 0.03312; AAV-BTNL9 0.03011; AAV-SRRD 0.00003. F) Select clusters of top protein-protein interactions (STRING database) of top 119 genes ranked by p-value that when knocked out increase the number of cells harboring TDP-43 $_{\Delta NLS}$ aggregates. Clusters generated with MCL clustering and excludes genes with no known connections and clusters with insignificant p-values. Clusters colored based on STRING annotated GO terms. Dashed line indicates noteworthy link of UBE3C to proteasomal degradation pathway G) Histograms of total TDP-43 $_{\Delta NLS}$ expression for AAV5 non-targeting control and two HSPA4 targeting sgRNAs. H) Fraction of cells with TDP-43 aggregates (y-axis) in each TDP-43 $_{\Delta NLS}$ expression bin (x-axis) for AAV5 non-targeting control and two HSPA4 targeting sgRNAs. Adjusted p-values i)bin 2500 AAV:HSPA4_C11 = 0.04711; AAV:HSPA4_C12 = 0.04054 ii) bin 3500 AAV:HSPA4_C11 = 0.03956; AAV:HSPA4_C12 = 0.04740 iii) AAV: HSPA4_C11 = 0.00341; AAV:HSPA4_C12 = 0.01095.

physically separate cells with and without inclusions by imaging sorted cells [30]. Because our aim was to find regulators of protein aggregation, we verified that the expression of our TDP-43 aggregation reporter did not induce cellular toxicity within the timeframe of our experiment (Figure A in S1 Text). This was consistent also for TDP-43 harboring ALS-associated mutations (Figure A in S1 Text).

We next performed a genome-wide CRISPR-Cas9 KO screen to identify gene modifiers of TDP-43 $_{\Delta NLS}$ protein inclusion formation in human cells following established protocols [31] (Fig 1C). To control for effects of transgene levels on aggregation, cells were sorted within a narrow range of transgene expression levels. We isolated cells from both the punctate and diffuse TDP-43 PulSA populations and used established protocols to isolate gDNA and amplify sgRNAs [32] to measure the abundance of sgRNA sequences between the two phenotypic groups. Amplicon sequencing data from the punctate and diffuse populations was analyzed to produce two values per gene: i) phenotype which estimates the phenotypic effect by taking the average of the two best performing sgRNA per gene, and ii) a p-value that calculates the significance of each gene using all sgRNAs [33] (Fig 1D). The screen was conducted in two replicates to ensure data reproducibility (Figure A in S1 Text), and the data were then aggregated for downstream analysis. To further validate the screen, we performed arrayed testing of top hits that increased or decreased TDP-43 foci. Compared to AAVS targeting control sgRNAs, two top hits that increased TDP-43 aggregation, UBE3C and HSPA4, followed the same trend observed in the screen and led to an increased number of cells harboring mClover3-TDP-43 $_{\Delta NLS}$ foci. Similarly, two top hits that decreased TDP-43 aggregation, SRRD and BTNL9, led to a decreased number of cells harboring mClover3-TDP-43 $_{\Delta NLS}$ foci (Fig 1E).

Top gene hits that increased inclusion formation when disrupted were highly enriched in proteostasis related functional annotations (Fig 1F and Figure A S1 Text). One of these genes, UBE3C, is a previously identified E3 ubiquitin ligase that modifies partially proteolyzed substrates [34]. Analysis of protein-protein interactions using the STRING database [35] of top hits (113 genes with p-value < 0.005 and LFC > 1) revealed a network of known proteostatic machinery, such as HSF-1, HSPA1B, several DNAJ proteins, and HSPA4 [4,36–39] (1D, Fig 1F and Figure A in S1 Text). A member of the HSP110 superfamily [40], HSPA4 is a nucleotide exchange factor that is a co-chaperone to HSP70 [41,42], the ubiquitous chaperone that aids in protein folding and disaggregation [43]. HSP70 chaperones RNA-binding deficient TDP-43 into liquid compartments [44], and the small heat shock protein HSPB1, which was found to prepare protein aggregates for disaggregation by HSP70, interacts with TDP-43 and slows the accumulation of insoluble TDP-43 [45]. Taken together, these findings suggest that HSPA4 may have a direct role in preventing the aggregation of mClover3-TDP-43 $_{\Delta NLS}$. To confirm that the observed effect of HSPA4 loss on mClover3-TDP-43 $_{\Delta NLS}$ foci formation was indeed due to effects on aggregation and not protein expression, we verified that HSPA4 disruption

did not cause a change in mClover3-TDP-43 $_{\Delta NLS}$ expression distribution by comparing the total expression levels between the cell lines (Fig 1G). We also show that the observed effect on aggregation was independent of expression levels by binning cells by expression levels (Fig 1H). A similar analysis on the other validated gene hits revealed a similar and consistent result (Figure B in S1 Text).

In addition to genes already associated with cellular proteostasis, our CRISPR KO screen uncovered several intriguing gene modifiers of mClover3-TDP-43 $_{\Delta NLS}$ inclusion formation. One such gene is METTL5 (Fig 1D and Figure A in S1 Text), a recently identified 18S rRNA m6A methyltransferase that plays a role in translation initiation and is a regulator of cellular stress responses [46–48]. A recent study found that METTL5 also strongly associates with RNA-binding proteins [49], suggesting a role in stress related translation regulation, the loss of which may contribute to the increase in mClover3-TDP-43 $_{\Delta NLS}$ inclusions we observed. Another interesting gene that, when depleted, increased the number of cells with mClover3-TDP-43 $_{\Delta NLS}$ aggregates was XPO4 (Fig 1D), a nuclear-export factor. To date, a number of studies have found that nuclear import factors act as chaperones for RNA-binding proteins by binding their nuclear localization signals to prevent or reverse their accumulation [50–52]. It is unclear if TDP-43 has an active Nuclear Export Signal (NES) [53], which might be required for XPO4 to exert chaperone activity, suggesting that the effect of XPO4 on mClover3-TDP-43 $_{\Delta NLS}$ aggregation may be indirect.

## APEX proximity labeling places SRRD in close proximity to intermediate filaments and parts of the endoplasmic reticulum

One of the top gene hits that decreased mClover3-TDP-43 $_{\Delta NLS}$ inclusions when perturbed was SRRD (Fig 1D), an uncharacterized gene that demonstrated a strong effect in both of our screen replicates (Figure A in S1 Text). SRRD was also recently identified as a modifier of SQSTM1 levels [54]. Together, these findings suggest a central and poorly studied role of this protein in cellular proteostasis. To study the cellular function of SRRD we first generated clonal SRRD KO HEK293T lines (Figure C in S1 Text). The clonal lines showed an effect on mClover3-TDP-43 $_{\Delta NLS}$ foci formation that was consistent with the screen and polyclonal KO validation (Figure C in S1 Text). Additionally, to verify that loss of SRRD gene product underlies the effect on aggregation and not an off-target effect of Cas9 KO, such as the generation of a truncated protein, we used CRISPRi to knock down SRRD and again observe reduced number of cells with mClover3-TDP-43 $_{\Delta NLS}$ aggregates (Figure C in S1 Text). SRRD displayed a similar effect on the aggregation of mClover3-FUS$_{P525L}$ (Figure D in S1 Text) suggesting that our observed effect is not specific to TDP-43 but might be unique to proteins similar to FUS and TDP-43 as both share a terminal prion-like domain with similar AA composition, enriched in uncharged polar residues & glycine. Interestingly, expression of mHTT(Q97) N-terminal fragment did not behave similarly, suggesting it is compartmentalized in these cells through distinct mechanisms (Figure D in S1 Text).

Next, to gain additional insight into the cellular role of SRRD we looked into its subcellular localization. Stable exogenous expression of SRRD-HA fusion displayed a cytoplasmic localization pattern that did not strongly colocalize with any obvious cytoplasmic markers, yet did display some overlap with an ER marker (Fig 2A). We also expressed SRRD-HA in HaLa cells, and observed a cytoplasmic localization pattern consistent with what we observed in 293Ts (Figure E in S1 Text). Finally, we expressed SRRD-HA in NGN2 induced neurons, and again observed predominantly soma localization of SRRD-HA (Figure E in S1 Text). To gain higher resolution information about the subcellular localization of SRRD we performed protein proximity labeling. We fused SRRD with an APEX2 enzyme which, in the presence of $H_2O_2$, will

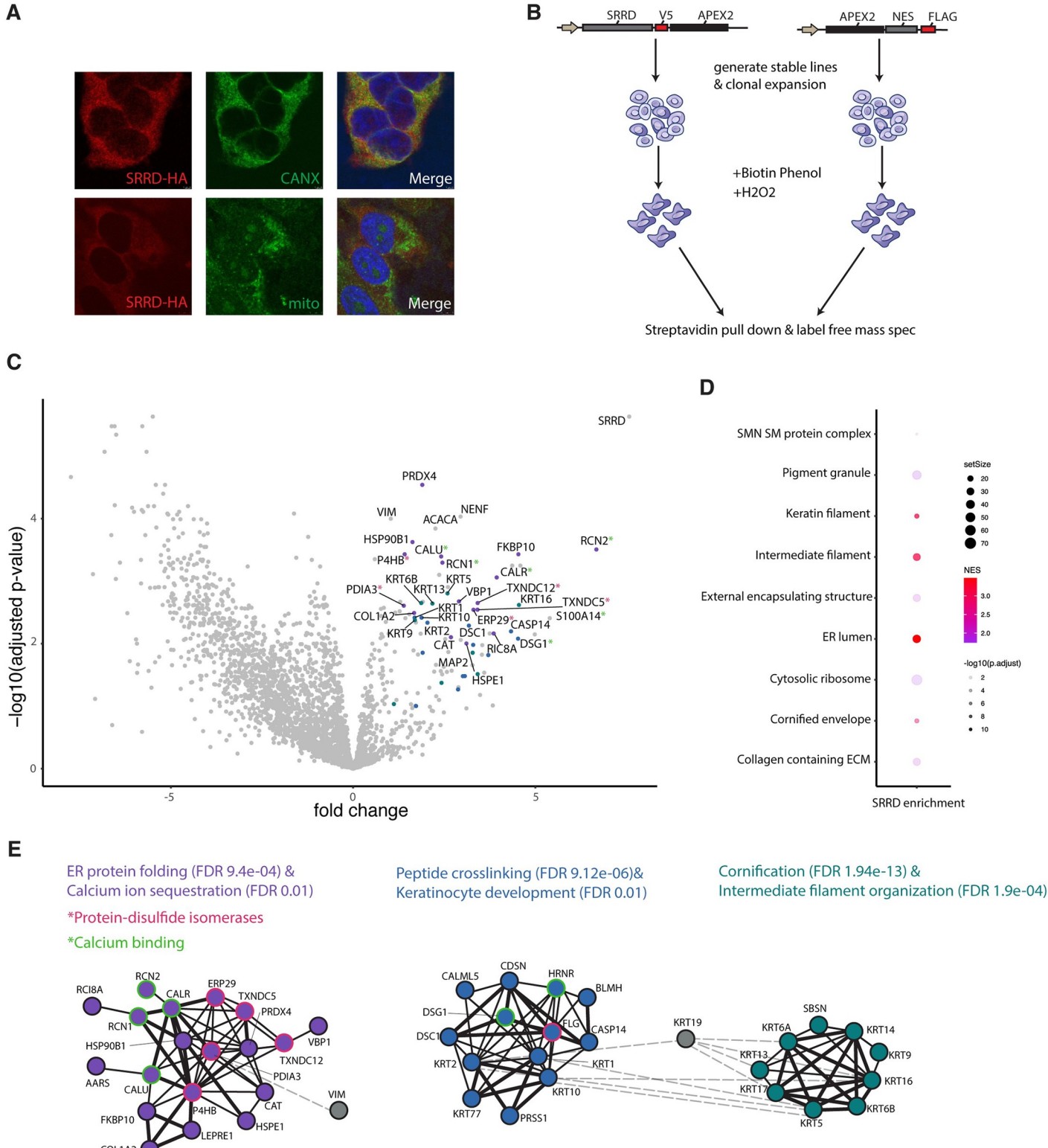

**Fig 2. APEX2 proximity labeling reveals SRRD in close proximity to intermediate filaments and regulators of IF oligomerization.** 1) HEK293Ts stably expressing SRRD-HA stained for HA and either CANX or mitochondria. B) Schematic of APEX2 proximity labeling experiment where APEX2 is fused to SRRD or to an NES control. C) Volcano plot of APEX2 proximity labeling mass spectrometry output, where fold change (x-axis) is plotted by significance (y-axis). Colored dots correspond to STRING clusters in 2E. (*) correspond to indicate functional annotations of interest highlighted in STRING cluster in 2E. D) Filtered GSEA (cellular

compartment) of SRRD-APEX2 dataset. E) Clustering of top protein-protein interactions (STRING database) of top 88 proteins ranked by fold change and p-value. Clusters generated with MCL clustering and excludes proteins with no known connections and clusters with insignificant p-values. Clusters colored based on STRING annotated GO terms and proteins with functional annotations of interested are highlighted as follows: Protein-disulfide isomerases circled in pink, proteins involved in calcium binding circled in orange.

label proteins in close proximity with biotin. Biotinylated proteins can then be pulled down with streptavidin beads and analyzed by mass spectrometry [55–57]. We used an APEX2 with an NES as a compartment control to rule out non-specific cytosolic biotinylation (Fig 2B). APEX2 activity was confirmed in both SRRD and control by showing that biotin and $H_2O_2$ treatment, in cells expressing APEX2, led to biotinylation of proteins (Figure F in S1 Text), and that biotinylation occurred in close proximity to those transgenes (Figure F in S1 Text). Finally, we performed a full labeling, pull down, and mass spectrometry experiment, where pulled down proteins were quantified using label free mass spectrometry in a Data Dependent Acquisition (DDA) manner and analyzed to identify differentially associated proteins (Fig 2C).

This analysis identified many proteins that were significantly more biotinylated in the presence of SRRD fused to APEX2, including SRRD as the top protein hit, serving as a technical experimental positive control (Fig 2C). To gain insight into the functional roles of the top protein hits we performed an unbiased Gene Set Enrichment Analysis (GSEA), using the Gene Ontology Cellular Compartment dataset, which highlighted IFs, including many keratin filaments, and ER lumen as the top enriched gene ontologies (Fig 2D). To gain functional insight into the very top proximal genes to SRRD, we performed a protein-protein interaction analysis using the STRING database followed by clustering (Fig 2E and Figure G in S1 Text). We found a large number of intermediate filament proteins, most notably keratin proteins, and the well characterized IF vimentin (VIM). Interestingly, many of the proteins identified, including ERP29, TXNDC5, TXNDC12, PDIA3, and P4HB, are annotated as protein disulfide-isomerases (PDI), a class of proteins that is responsible for the formation and breakage of disulfide bonds. Another prevalent molecular function within the APEX hit proteins was calcium binding, including S100A proteins, RCN1, RCN2, CALR, CALU, and DSG1.

IFs are a large class of proteins that comprise one of the three major branches of the cytoskeleton. IF proteins can form a variety of cellular structures, ranging from rigid extracellular appendages such as hair and nail but also essential and dynamic cellular networks that scaffold organelles, sense and respond to mechanical stimuli, and respond dynamically to intracellular stress [58–60]. Unlike actin and microtubules, regulation of the dynamic assembly and disassembly of IFs is very understudied and poorly understood. What has been shown is that keratin heterodimers can be linked by disulfide bonds when they are in oxidizing environments [28,61] like the perinuclear space [62], and disulfide bonds increase when $Ca^{2+}$ concentrations increase [28]. Without these disulfide bonds normal filament elongation, response to mechanical signals, and the generation of a perinuclear keratin network are abrogated [61]. The detection of SRRD in close proximity to IFs, PDIs, and calcium binding proteins suggests that it may play a role in the regulation of the spatial organization of the IF network.

## Loss of SRRD leads to dysregulation of intermediate filaments

We next tested the hypothesis that loss of functional SRRD in the dynamic regulation of the IF network underlies the reduced foci formation we observed in our screen. To do this, we examined VIM organization in wild type HEK293Ts, SRRD clonal KO HEK293Ts, and SRRD clonal KO with SRRD stable reintroduction (SRRD rescue) HEK293T cell lines. We observed VIM organized into smooth bundles and projections in the WT and SRRD rescue cells, while

VIM in the SRRD KO cells appeared fragmented and disorganized (Fig 3A and Figure H in S1 Text). In light of this observed disorganization, we tested if loss of SRRD would have more general effects on the levels of various IF proteins. To address this we compared protein abundance between wild type and SRRD clonal KO HEK293Ts using label free mass spectroscopy. Proteomic data was collected in a Data Independent Acquisition (DIA) manner and statistically analyzed to identify differentially expressed proteins (Fig 3B). STRING and GSEA, using the Gene Ontology Cellular Compartment dataset, revealed a reduction in several intermediate filament (IF) proteins and IF-associated proteins (Fig 3B–3D and Figure I in S1 Text) supporting the hypothesis that loss of SRRD leads to dysregulation of this branch of the cytoskeletal network. To further establish the connection between IF dynamics and the formation of TDP-43 inclusions, we treated cells with the IF inhibitor (-) Epigallocatechin gallate, a drug that binds VIM and leads to altered levels of both VIM and cytokeratins [63,64]. Expression of TDP-43 aggregation reporter in (-) Epigallocatechin gallate treated cells lead to a reduced number of cells harboring TDP-43 PulSA aggregates compared to DMSO control (Figure H in S1 Text), supporting our hypothesis that SRRD affects TDP-43 aggregation via an effect on IF organization.

To see if this effect on the cytoskeleton will also be observed in other cell types, we turned to an inducible neuron model, as neurons require a highly organized cytoskeleton to maintain cell polarity and function [65–68]. While VIM is only expressed in immature neurons [69], neurons also express INA, which was one of the top proteins that we found depleted upon SRRD KO in 293Ts. Human induced pluripotent stem cells were differentiated into neurons using Ngn2 induction [70], and SRRD was knocked down via CRISPRi. Imaging analysis of SRRD CRISPRi neurons compared to control revealed increased MAP2 and INA signal in the soma, and reduced MAP2 and INA staining in dendritic and axonal projections, respectively (Fig 3E and 3F). We confirmed that this was not due to a reduction in absolute levels of these proteins by western blot (Figure J in S1 Text).

## Loss of SRRD affects aggresome formation and composition

As excess of unfolded proteins in eukaryotic cells are sequestered into specialized compartments, we hypothesized that loss of SRRD may alter the formation of aggresomes, perinuclear deposits of unfolded proteins that are enriched in degradation machinery and encapsulated in IFs like VIM [6–10,11,12,71]. TDP-43 is known to be recruited to aggresomes during prolonged stress [72] and recent work has shown that VIM is critical for localizing degradation machinery to aggresomes [73]. To address this, we tested aggresome formation in WT HEK293Ts, SRRD clonal KO HEK293Ts, and SRRD rescue HEK293Ts following treatment with 5μM MG132, a proteasome inhibitor that induces aggresome formation [8,73,74], or DMSO control. We defined aggresomes as nuclear deforming deposits that showed both VIM cage and HDAC6 staining. Interestingly, WT cells showed clear aggresome formation, a drastic reduction in aggresome formation was observed in SRRD KO with partial rescue following the reintroduction of SRRD (Fig 4A and 4B and Figure K in S1 Text).

After confirming that loss of SRRD reduces aggresome formation during general proteotoxic stress, we tested if SRRD depletion reduces mClover3-TDP-43 $_{\Delta NLS}$ inclusion formation via a similar mechanism. Perturbed recruitment of mClover3-TDP-43 $_{\Delta NLS}$ to aggresomes following loss of SRRD would explain our initial screen results where SRRD depletion leads to fewer detectable mClover3-TDP-43 $_{\Delta NLS}$ inclusions. Using the same cell lines, WT HEK293Ts, SRRD clonal KO HEK293Ts, and SRRD rescue HEK293Ts, we expressed the mClover3-TDP-43 $_{\Delta NLS}$ reporter, and examined the localization of exogenous TDP-43 with VIM. In WT cells, we observe strong VIM cage formation around most TDP-43 foci, which is

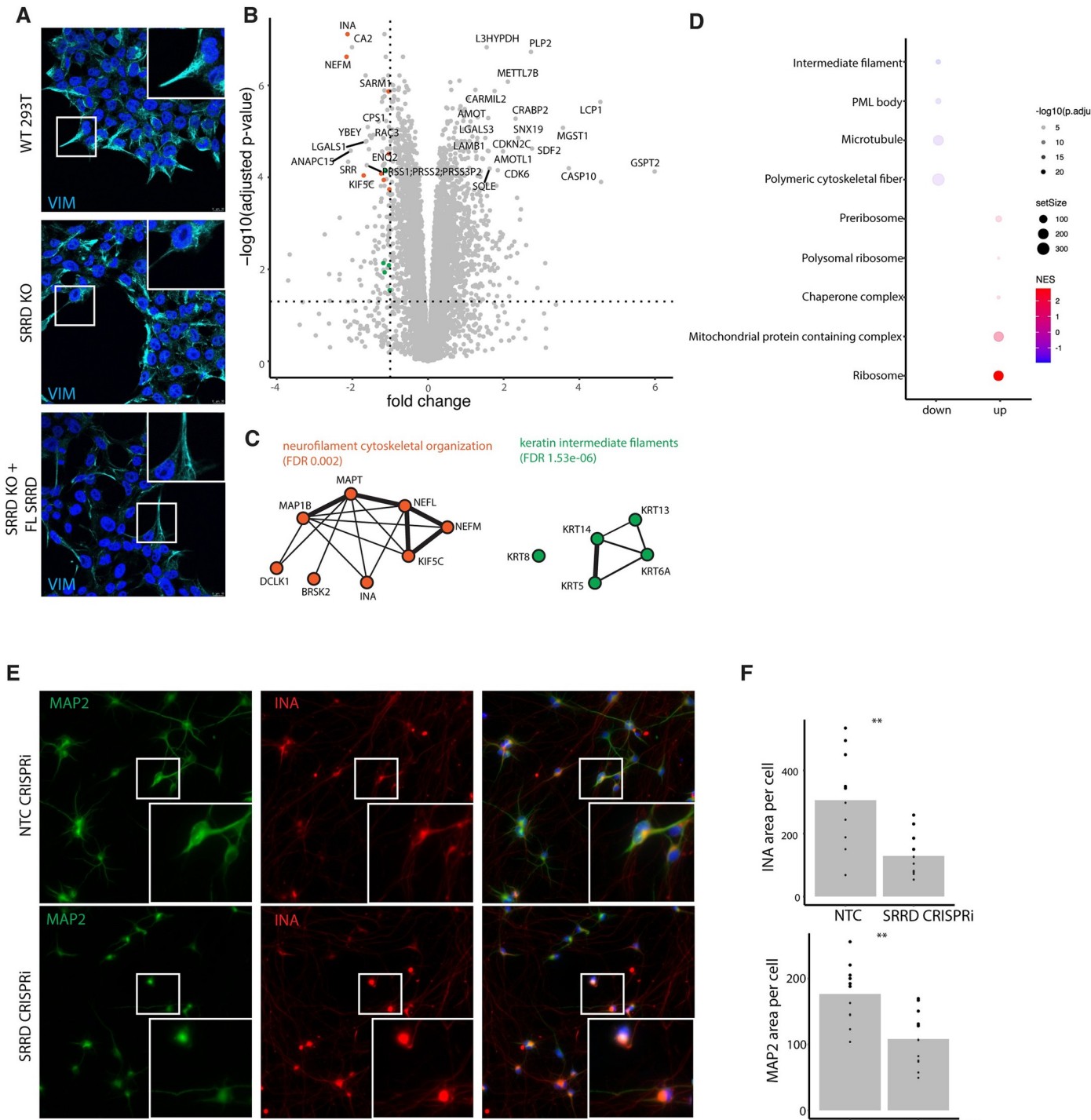

**Fig 3. Loss of SRRD results in disorganized and downregulated IFs.** A) Confocal images of indicated cells lines stained for VIM. B) Volcano plot of quantitative proteomics experiment comparing SRRD clonal KO HEK293Ts to WT HEK293Ts where fold change (x-axis) is plotted by significance (y-axis). Horizontal dashed line represents adjusted p-value cutoff of 0.05, vertical line represents fold change of -1. Orange and green colored dots correspond to STRING clusters in 3C. C) Select clusters of top depleted proteins in SRRD KO (STRING database) ranked by fold change and p-value. Clusters generated with MCL clustering and excludes proteins with insignificant p-values. Clusters colored based on STRING annotated GO terms. D) Filtered GSEA (cellular compartment) of quantitative proteomics dataset. E) Representative images of NGN2 neurons transduced with SRRD CRISPRi sgRNA or non-targeting control, stained for MAP2 and INA. F) Quantification of the area per cell covered by INA and MAP2 signal in SRRD CRISPRi and NTC control NGN2 neurons.

evident by microscopy and also by line intensity plots showing strong VIM signal flanking TDP-43 (Fig 4C-F and Figure K in S1 Text). Strikingly, when SRRD is depleted, VIM cage formation around mClover3-TDP-43 $_{\Delta NLS}$ is all but lost (Fig 4C-F and Figure K in S1 Text). Moreover, we found a more dispersed mClover3-TDP-43 $_{\Delta NLS}$ signal in the SRRD KO cells with reduced intensity, longer tails of the distribution, and higher standard deviation compared to WT cells (Figs 4D and 4E). When full length SRRD is re-expressed, we see a partial rescue of VIM cages and less diffuse TDP-43 signal (Figs 4C-F and Figure K in S1 Text). From these data, we conclude that without SRRD, VIM cages fail to form, which is accompanied by reduced accumulation of mClover3-TDP-43 $_{\Delta NLS}$ into cytoplasmic foci. Despite the reduced number of aggresomes and lack of VIM, the mClover3-TDP-43 $_{\Delta NLS}$ aggresomes that form are positive for HDAC6 (Figure L in S1 Text), suggesting that SRRD plays an important role in the efficiency of aggresome formation (Fig 4A) and the composition of the resulting aggresomes, but is not essential for HDAC6-mediated trafficking of aggresome cargo.

Next, we tested if loss of SRRD might have an effect on the localization of additional aggresome components, like SQSTM1 (p62) [71,75]. SQSTM1 was of special interest to us, due to a previous screen that found SRRD KO resulted in increased levels of SQSTM1 [54]. Given that SRRD was previously established to regulate SQSTM1 and we have shown that SRRD affects aggresome formation, we hypothesized that depletion of SRRD may reduce recruitment of SQSTM1 into aggresomes. We thus examined localizations of SQSTM1 in WT HEK293Ts, SRRD clonal KO HEK293Ts, and SRRD rescue HEK293Ts treated with 5μM MG132 or DMSO control. Interestingly, in unstressed conditions, we found that SRRD KO cells have a greater number of large SQSTM1 puncta per cell, suggesting changes in autophagic flux even in the homeostatic state (Figure L in S1 Text). Under stress, SQSTM1 accumulates into HDAC6+ aggresomes of all cell lines (Fig 4G), but the intensity of SQSTM1 inside aggresomes is reduced in SRRD KO cells (Figs 4G-I).

## SRRD localizes to unfolded protein inclusions during stress

We next asked if the localization of SRRD changes during proteotoxic stress, which may suggest a direct role in the compartmentalization of unfolded proteins. To test if SRRD gains additional interactions during stress, we used the same SRRD APEX2 fusion as previously described (Fig 2B) and performed a full labeling, pull down, and mass spectrometry experiment comparing cells treated with MG132 (to induce aggresome formation) or DMSO control (Fig 5A). Pulled down proteins were quantified using label free mass spectroscopy in a Data Dependent Acquisition (DDA) manner and analyzed to identify differentially associated proteins (Fig 5B). STRING analysis and (GSEA) using the CORUM protein complex dataset revealed that in stress conditions SRRD gains proximity to components of the proteasome, the centrosome, and proteins involved in protein folding (Fig 5C, 5D and Figures E in S1 Text).

The stress specific association of SRRD with the centrosome and protein degradation machinery suggests that SRRD localizes to aggresomes, as they are formed at the Microtubule Organizing Center (MTOC), are positive for centrosomal components, and are enriched in protein degradation machinery [8,74]. Indeed, SRRD fused to mRuby3 relocalizes to puncta within aggresomes that are both VIM and HDAC6 positive following proteasome inhibition (Fig 5E and Figure N in S1 Text). We further verified that APEX2 fused SRRD also relocalizes to perinuclear bodies following MG132 treatment (Figure N in S1 Text). Similarly, when aggresomes were formed by exogenous expression of mClover3-TDP-43 $_{\Delta NLS}$, we observed a strong colocalization between SRRD and mClover3-TDP-43 $_{\Delta NLS}$ foci when compared to mClover3 control suggesting either a direct interaction with unfolded TDP-43 or co-sequestration by a shared mechanism (Fig 5F and 5G and Figure N in S1 Text).

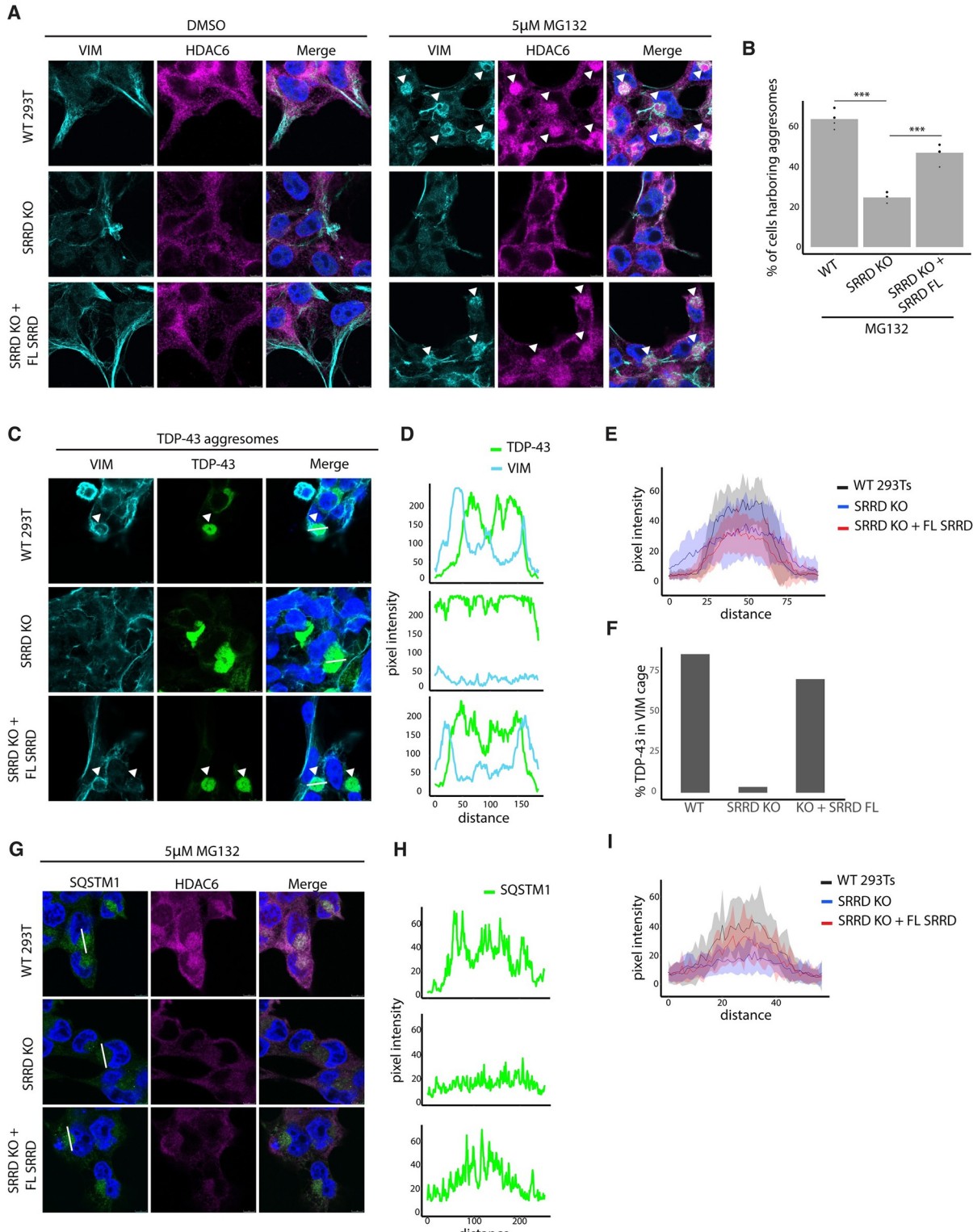

**Fig 4. SRRD regulates efficient assembly of aggresomes.** A) WT, SRRD clonal KO, and SRRD clonal KO + SRRD-mRuby3 293Ts treated for 16hrs with 5μM MG132 or DMSO control and stained for VIM and HDAC6. B) Quantification of the percentage of cells harboring aggresomes (perinuclear, HDAC6+, VIM cage+) after MG132 treatment. Dots indicate replicate wells treated, stained, and imaged in parallel. n = 4 replicates, one way anova with Tukey HSD test. Adjusted p-values WT:SRRD KO = 0; KO:rescue = 2.71e-09. C) WT, SRRD clonal KO, and SRRD clonal KO + SRRD-mRuby3 293Ts transfected with mClover3-TDP-43 $_{\Delta NLS}$ and stained for VIM. D) Line intensity plots of TDP-

43 $_{\Delta NLS}$ (green) and VIM (blue) signals corresponding to white lines drawn in 3C. E) Line intensity drawings aggregating TDP-43 $_{\Delta NLS}$ intensity data from WT, SRRD clonal KO, and SRRD clonal KO + SRRD-mRuby3 293Ts. Solid line indicates average value at each point, and shaded areas represent the standard deviation. F) Quantification of the percentage of TDP-43 $_{\Delta NLS}$ aggregates that have at least a partial VIM cage surrounding it in WT, SRRD clonal KO, and SRRD clonal KO + SRRD-mRuby3 293Ts transfected with mClover3-TDP-43 $_{\Delta NLS}$ and stained for VIM. G) WT, SRRD clonal KO, and SRRD clonal KO + SRRD-mRuby3 293Ts treated for 16hrs with 5μM MG132 and stained for SQSTM1 and HDAC6. H) Line intensity plots of SQSTM1 corresponding to white lines drawn in 3G. I) Line intensity drawings aggregating SQSTM1 data from WT, SRRD clonal KO, and SRRD clonal KO + SRRD-mRuby3 293Ts. Solid line indicates average value at each point, and shaded areas represent the standard deviation.

## SRRD low complexity N terminal domain is sufficient for localization with aggresomes and unfolded proteins

We next examined the predicted structure and domains of SRRD, using Eukaryotic Linear Motif [76] and AlphaFold prediction [77,78] respectively, to see if this would provide insight into it function and interactions. Beyond the SRR1-like domain for which the protein is named, SRRD also contains a low complexity N-terminal domain (NTD) spanning amino acids 1–50 that is rich in arginine (~30%) and alanine (~35%) residues, making it positively charged, hydrophobic, and predicted to be unstructured (Fig 6A and Figure O in S1 Text). Expression of the NTD alone was sufficient to induce co-localization (Fig 6B) while SRRD lacking the NTD was unstable and showed diffuse localization (Figure O in S1 Text). We next generated truncations of SRRD tagged with mRuby3 and delivered them with mClover3-TDP-43 $_{\Delta NLS}$ into HEK293Ts. When we quantified the percentage of cells harboring TDP-43 foci after delivery of multiple SRRD truncations. We found that the truncations that retain the N terminal domain significantly reduced the number of cells harboring TDP-43 foci (Fig 6C), potentially by binding TDP-43 and either inhibiting interactions with other TDP-43 mono-mers or blocking interactions with protein machinery that brings TDP-43 into aggresomes.

As an additional model to indirectly test the effect of SRRD expression on aggregation prone proteins, we turned to a yeast model of protein toxicity where expression of non-native proteins that are prone to aggregation, like TDP-43, FUS, and alpha-synuclein, leads to toxicity and reduced growth [14,30,79,80]. Expression of SRRD led to a significant rescue of growth for several aggregation prone proteins (Fig 6D and 6E and Figure O in S1 Text) when com-pared to HSP104-A503S, an engineered disaggregase which was optimized to rescue toxicity in this model [80] (Fig 6D and 6E and Figure O in S1 Text). Similar to SRRD truncations in mammalian cells, the NTD alone was sufficient to observe reduced toxicity while longer trun-cations that still contained the NTD did not show any significant effect Fig 6F. Toxicity in these models is likely due to toxic interactions between the human aggregation prone proteins to essential yeast proteins [81]. These results suggest a possible direct interaction between unfolded proteins and SRRD NTD, yet more work would be required to investigate if this interaction has a cellular or it reflects co-sequestration through a shared mechanism.

## Discussion

Despite the prevalence of pathogenic protein aggregates in an array of human diseases, the mechanisms by which these protein inclusions form remains poorly understood [4,5,19,82]. Here, we describe a novel screening approach that couples a genome-wide CRISPR-Cas9 KO screen to a PulSA-amenable protein aggregation reporter using TDP-43 as our model protein. The vast majority of our top screen hits, especially those that increase protein inclusion forma-tion when depleted, are associated with the proteostasis network. This demonstrates that screening directly for protein aggregation, as opposed to using toxicity as a cellular phenotype, is a powerful strategy to identify gene modifiers that are directly related to protein inclusion

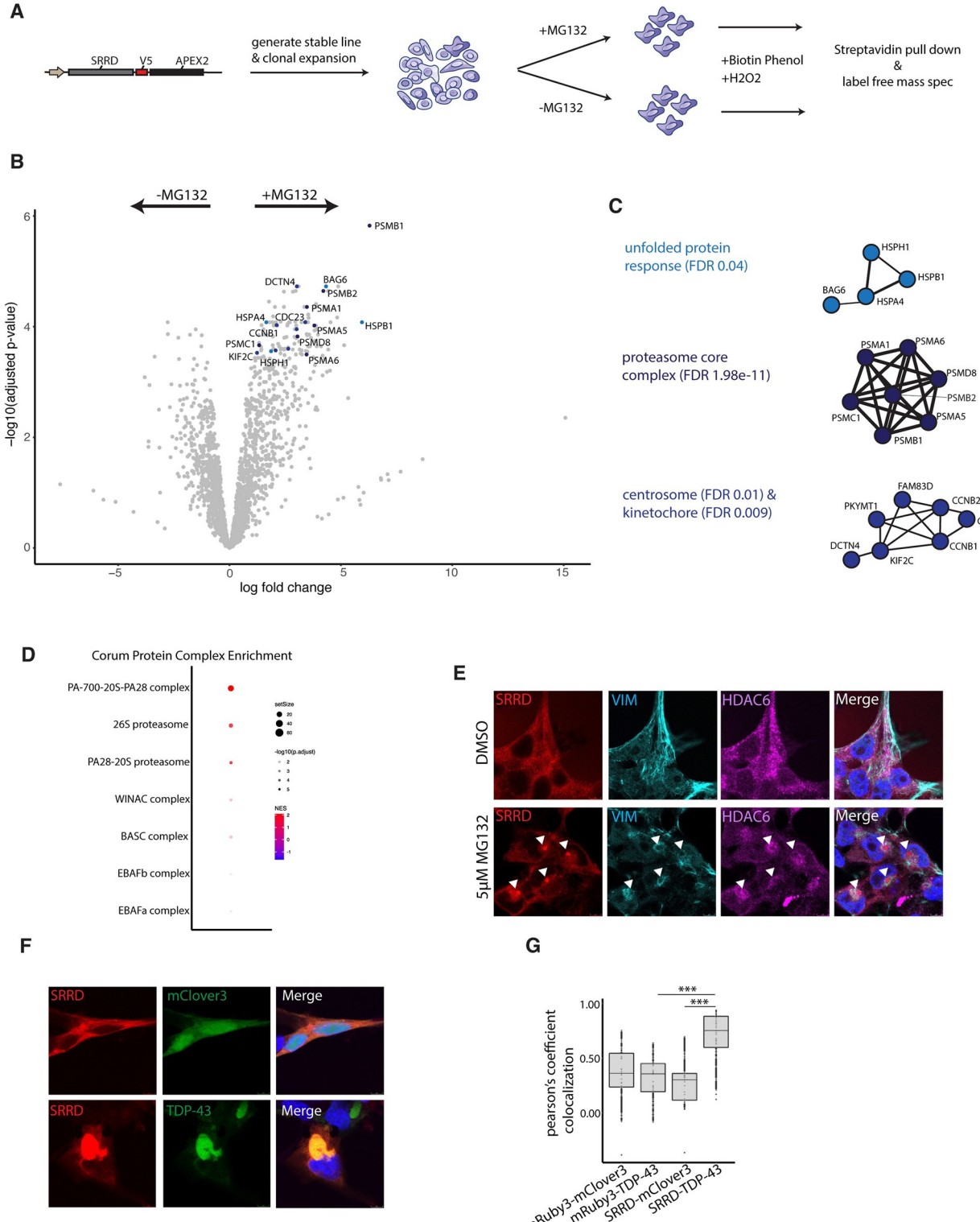

**Fig 5. SRRD localizes to aggresomes following cellular proteotoxic stress.** A) Schematic of APEX2 proximity ligation assay with SRRD-V5-APEX2 clonal lines. B) Volcano plot of APEX2 proximity labeling mass spectrometry output, where fold change (x-axis) is plotted by significance (y-axis). Arrows indicate enrichment in +/- MG232 conditions, and each dot represents a protein. Colored dots correspond to STRING clusters in 2I. C) Clustering of top protein-protein interactions (STRING database) of top 100 proteins ranked by fold change and p-value associated with +MG132 (blue) and top associated proteins associated with -MG132 (red) conditions. Clusters generated with MCL clustering and excludes

proteins with no known connections and clusters with insignificant p-values. Clusters colored based on STRING annotated GO terms. D) Enrichment analysis of ranked proteins after differential expression analysis of APEX +/- MG132 proximity labeling experiment using CORUM protein complex database. E) 293Ts expressing SRRD-mRuby3 treated with either 5μM MG132 or DMSO control for 16hrs, fixed and stained for VIM and HDAC6. F) 293Ts stably expressing SRRD-mRuby3 transfected with mClover3 control or mClover3-TDP-43 $_{\Delta NLS}$. G) Colocalization of mRuby3 and mClover3 in indicated protein pairs, measured in FIJI using Pearson's correlation coefficient. Each dot represents correlation coefficient calculated for a single cell, boxes indicate median, upper, and lower quartiles. T-test p-values: SRRD-mClover:SRRD-TDP-43 = 1.055e-08; mRuby3-TDP-43:SRRD-TDP-43 = 7.305e-10.

formation. Several of our top gene hits were previously associated with protein aggregation, such as UBE3C [34], HSF1, and several chaperones including HSPA4, HSPA1B, DNAJC7, and DNAJC12. The discovery of chaperones that act directly on TDP-43 aggregation is of great interest, as it helps to better understand the molecular network that maintains TDP-43 in its functional form [45] and may eventually form the basis for novel therapeutic development.

Other less expected hits include XPO4 and METTL5. While recent discoveries suggest that nuclear-transport factors are capable of chaperone-like activity [50–52], limited data currently exists about exportins [83], yet more work will be required to show if this is a direct effect and not through changes in the composition of the cytoplasm due to loss of XPO4. Regarding METTL5, a recently identified RNA methyltransferase, it is known that mRNA modifications have been shown to affect aggregation by affecting the RNA-binding activity of TDP-43 and other RBPs [84–86]. The mechanism by which METTL5 loss affects aggregation might be different though, as the RNA clients of METTL5 were recently shown to be rRNA molecules that, when modified, will affect the translation of stress response proteins [46,47]. More work is needed to understand if METTL5 also acts on known TDP-43 mRNA clients, or if through modifications of translation machinery METTL5 affects protein burden and general proteostasis instead.

Given the setup of our screen, we expected to find multiple gene hits that act to oppose protein aggregation. More surprising was to find strong gene modifiers that show the opposite effect, suggesting that they are involved in active compartmentalization of unfolded proteins. While it is well established that appropriate nuclear localization of TDP-43 is essential [87,88], it is less clear how compartmentalization of TDP-43 in the cytosol contributes to cell viability. Many studies suggest that compartmentalization of TDP-43 into ribonuclear granules seeds pathological aggregates and contributes to loss-of-function toxicity [89,90]. To the contrary, other groups have shown that active compartmentalization of TDP-43 can also be protective against toxicity [84,91,92]. We identified SRRD as a top gene hit that when depleted reduced TDP-43 aggregation, suggesting a role for TDP-43 compartmentalization when present. SRRD was also identified in another recent genome-wide screen that looked for modifiers of SQSTM1 levels [54]. The fact that SRRD was identified in these two unrelated, unbiased genome-wide screens suggests that it is an important regulator of cellular proteostasis with a poorly understood and understudied molecular function. Thus, we decided to focus our follow up studies on this protein.

Our APEX data suggest that SRRD resides in close proximity to IF proteins, specifically many keratin genes and VIM, and to several proteins with molecular functions that are either PDIs or calcium binding. The dynamic assembly and disassembly of the keratin network and likely other IF proteins depends on formation and breakage of disulfide bonds in a manner that is strongly affected by calcium levels [28]. This suggests that SRRD may have a direct role in the spatial arrangement of the IF network. Indeed, we find impaired assembly of VIM fibers in homeostatic conditions and a complete lack of VIM cages in aggresomes following proteotoxic stress. Reintroduction of SRRD was able to partially rescue this effect. The partial effect can probably be explained by the unmatching expression levels between the KO and rescue as

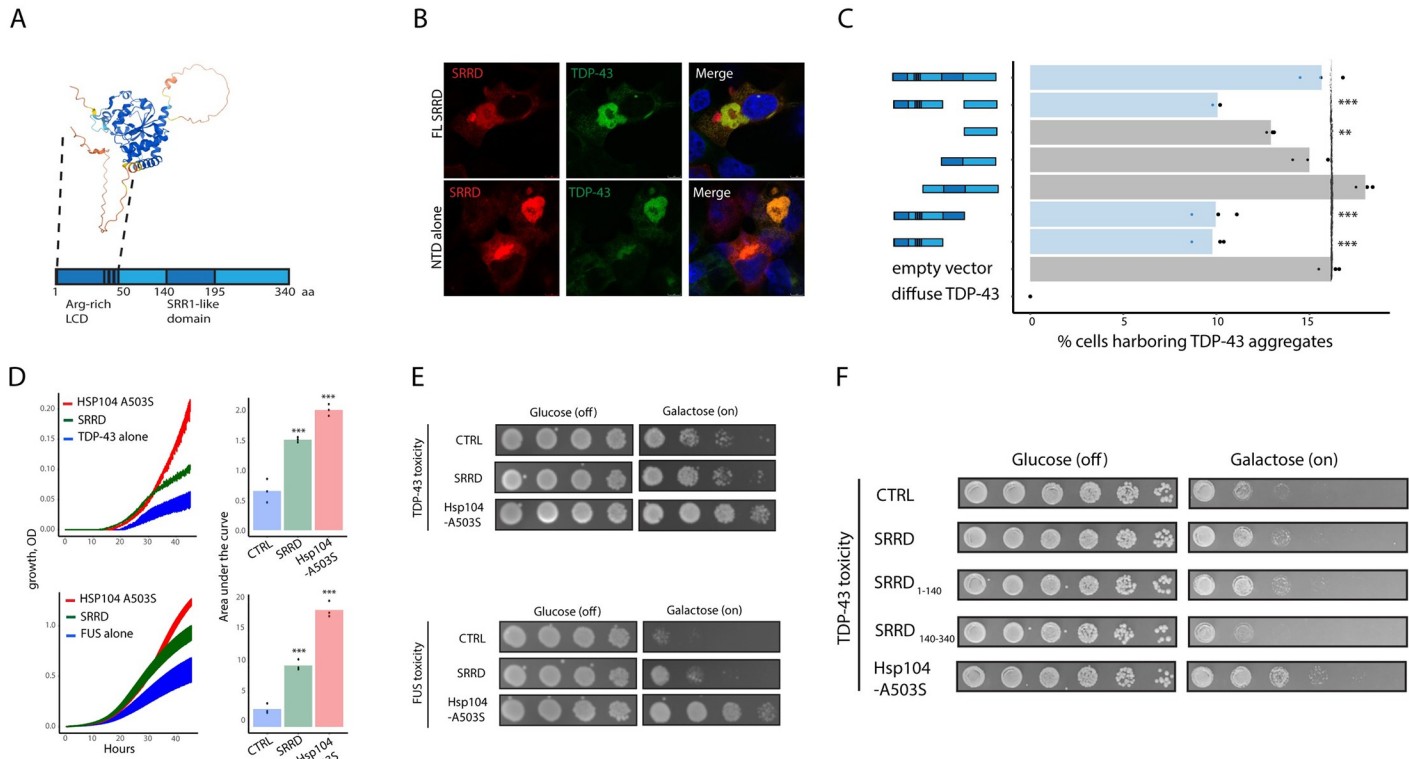

**Fig 6. SRRD N-terminal low complexity is sufficient to rescue protein toxicity and inclusion formation in orthogonal models of protein toxicity and aggregation.** A) AlphaFold predicted structure of SRRD (top) and schematic of SRRD indicating predicted domains (ELM) with amino acid numbers indicating the size of each domain. Dashed lines indicate N-terminal low complexity domain on both schematic and AlphaFold structure. B) 293Ts stably expressing SRRD-mRuby3 or NTD-mRuby3 (1-140aa) transfected with mClover3-TDP-43 $_{\Delta NLS}$. C) 293Ts stably expressing SRRD-mRuby3 or NTD-mRuby3 (1-140aa) transfected with mClover3-TDP-43 $_{\Delta NLS}$. F) FACS analyzed 293Ts co-transfected with mClover3-TDP-43 $_{\Delta NLS}$ and indicated truncations of SRRD (P2A-mRuby3 empty vector as expression control). Percentage of cells harboring TDP-43 $_{\Delta NLS}$ aggregates plotted for each SRRD truncation. n = 3 replicates, one way anova with Tukey HSD test. Vector:0–150 p-value = 0.00000; vector:0–194 p-value = 0.00000; vector:50–340 p-value = 0.19792; vector:150–340 p-value = 0.70249; vector:195–340 p-value = 0.00387; vector:-SRR1 p-value = 0.00000; vector:FL p-value = 0.99494. Bars representing SRRD truncations harboring the N-terminal domain are colored blue, bars representing SRRD truncations lacking the N-terminal domain are colored grey. D) Yeast growth over time, measured by optical density. Yeast expressed TDP-43 or FUS alone (blue lines), in combination with HSP104-A503S disaggregase (red lines), or in combination with SRRD (green lines). And area under the curve of each growth curve. Adjusted p-values: Hsp104-A503S rescue of TDP-43 toxicity = 0.00003; SRRD rescue of TDP-43 toxicity = 0.00045; Hsp104-A503S rescue of FUS toxicity = 0.00000; SRRD rescue of FUS toxicity = 0.00035. E) Yeast spotting assay where indicated transgene expression is off under glucose, on under galactose. F) Yeast spotting of TDP-43 toxicity where TDP-43 and indicated SRRD truncation or Hsp104-A503S expression is off under glucose, on under galactose.

exogenous expression is likely to be much higher than endogenous expression which might affect SRRD function. Unlike microtubules (MT), which have been extensively studied over the years, the regulation of IF spatial cellular dynamics is poorly understood and understudied. Interactions between SRRD and unfolded IF and other proteins, likely occurs only during assembly and disassembly as SRRD localization pattern does not show fibrillar pattern that would be expected from a protein that is associated with assembled IF. We have attempted to further study the potential interactions between SRRD and IF proteins using biochemical approaches (e.g. APMS) yet none of these approaches were successful suggesting that gaining better understanding of such chaperone-like weak interactions would require the use of in-vitro approaches [93,94]. Whether SRRD is involved in the regulatory pathways of the PDIs or calcium binders, facilitating localization of IFs and regulatory proteins, or monitoring protein folding state is unclear at this time. Follow up work, including higher resolution mapping of physical protein-protein interactions and closer dynamic imaging would help decipher the

precise molecular mechanisms which will significantly advance our understanding of IF dynamics and regulation.

Our findings strongly highlight the connection between IF spatial regulation and cellular proteostasis. The most explored aspect of this relationship is the role of VIM in aggresome formation. It has also been suggested that VIM recruitment to the periphery of aggresomes facilitates the localization of proteasomes, chaperones, and ribosomes into aggresomes [73] which might explain the reduced accumulation of mClover3-TDP-43 $_{\Delta NLS}$, HDAC6, and SQSTM1 into aggresomes which we observe following SRRD loss (Figs 4C-H).

Given that SRRD localizes to aggresomes and unfolded proteins, and seems to affects toxicity and aggregation via its N-terminal low complexity domain, it is tempting to postulate that SRRD depletion is also affecting aggresome composition through direct interactions with unfolded proteins. Low complexity domains are well established to mediate protein-protein interactions [82,95,96], and TDP-43 also harbors a C-terminal low complexity domain that facilitates its fibrillization [82,84]. One hypothesis could be that the low complexity domain of SRRD is able to interact with unfolded proteins to recruit them into aggresomes. An alternative hypothesis could be that SRRD is acting as a chaperone that facilitates Intermediate filaments assembly, which are composed of fibrillar proteins prone to aggregation [97] and it is simply being co-sequestered into aggresomes together with other unfolded proteins. Much more work is required to distinguish between these two hypotheses and better understand the precise underlying molecular mechanisms.

While it is clear that TDP-43 mislocalization and aggregation is associated with long term neuronal toxicity and disease, the precise mechanisms through which this occurs are still unclear, thus it is hard to make clear suggestions if SRRD should be up or down regulated and more work is required to know if IF network modulation is a viable therapeutic strategy. Furthermore, all published neuronal models of TDP-43 aggregation, which we are aware of, rely on prolonged exogenous stress that leads to stress granule localization [98]. Without a robust neuronal model of TDP-43 pathology, it will be exceedingly difficult to determine the therapeutic value of altering SRRD levels, or any other hits identified in cell line models, to promote or eliminate TDP-43 aggregates. It is worth noting that while we use toxicity as a readout in our yeast experiments, it is only used to provide more support to a potential chaperone-like interaction between the two proteins, given that toxicity in the yeast system is induced by exposed unfolded proteins disrupting essential yeast proteins [81], and that assessing such interactions in cells or using biochemical means is challenging.

Lastly, SRRD seems to have a major impact on cellular proteostasis even without inducing acute proteotoxic stress as evident by its identification of a top gene modifier of p62 levels [54] and our observation of p62 positive puncta in homeostatic cellular conditions (Fig 4G and Figure M in S1 Text). Further investigation may reveal an important pathway in which IF proteins are involved in the maintenance of homeostatic cellular proteostasis, and that aggresome formation is the extreme manifestation of such pathway under very severe proteotoxic stress.

## Methods

### Cell culture

**Maintenance.** Human 293Ts (ATCC CRL-3216) were maintained in DMEM (Gibco #11995–065) with 10% FBS (Life Sciences) and 1% NEAA (Gibco #11140076). Cells were grown at 37C with 5% $CO_2$ to maintain physiological pH. Cells were tested for mycoplasma contamination after each thaw and before experimental use.

**Transfection.** Cells were plated on a 0.1% gelatin coated surfaces such that they would be 75% confluent at the time of transfection. Between 30min and 1hr prior to transfection media

was changed to DMEM, 10% FBS, 1% NEAA, plus 1% HEPES (Invitrogen), and added at 70% normal volume. For TDP-43 expression experiments, cells were transfected with a total of 2ug DNA per well of a 6-well plate or 1ug DNA per well of a 12-well plate, using a 1:3 ratio of total DNA to PEI (polyethylenimine HCl MAX transfection reagent (Polysciences, Inc)). Media was changed 6hrs after transfection to DMEM, 10% FBS, 1% NEAA. For viruses used in iPSCs, the media post-transfection was switched to mTeSR1 media. In experiments using TDP-43 expression vector, 1ug/ml Doxycycline (Hyclate) Hydrochloride (Sigma #9891) was added to media. For aggregation reporter validation and post-screen transfection-based experiments, transfected cells were lifted 24hrs post doxycycline induction and analyzed by FACS (FACSAria Fusion BD).

**Lentiviral generation.** Human 293Ts prepared for transfection as described above. Lentivirus was prepared for individual sgRNAs and plasmids by co-transfecting 293Ts with 1.06ug pMDLG (Addgene #12251), 0.57 pMD2G (Addgene #12259), 0.4ug pRSV-Rev (Addgene #12253), 1.06ug plasmid to be packaged, and a 1:3 ratio of total DNA to PEI into individual wells of a 6-well plate. Supernatant was collected 48hrs later, filtered through 0.45μm filter, and stored at -80 until use. Lentivirus was then thawed on ice before used for transduction. Lentivirus prepared for pooled libraries was scaled up by co-transfecting 293Ts seeded on 0.1% gelatin coated 15cm plates with 13.25ug pMDLG, 7.2ug pMD2G, 5ug pRSV-Rev, 20ug of pooled Brunello library (Addgene #73178) and 136ul PEI per 15cm plate. Supernatant was collected and replaced with DMEM + 10% FBS and 1% NEAA at 24hrs, and collected a final time at 48hrs post transfection. Supernatant was filtered, aliquoted, and stored at -80C until use.

## Induced neuron culture

**iPSC maintenance.** CRISPRi-i[3]N iPSCs (male WTC11 background; gifted by Dr. Michael Ward, NIH), described previously [99], were cultured in mTesR1 medium (STEMCELL Tech; Cat. No. 85850) on 6-well cell culture plates coated with hESC-Qualified, LDEV-Free, Matrigel Matrix (Corning; Cat. No. 354277) diluted according to manufacturer's lot number recommendation. mTesR1 medium was replaced every day until 80%–90% confluent, when cells were then passaged using Versene (GIBCO/Thermo Fisher Scientific; Cat. No. 15040066). Briefly, media was aspirated and then cells were washed 1x with DPBS and then incubated with Versene at 37C for at least 5min. The Versene was then aspirated, and the cells lifted by washing the well with fresh mTesR1 medium and gently scraping if needed. Colonies were broken up by gently triturating the cell mixture 3x before transferring the cells to a new Matrigel-coated plate at desired concentration.

**Neuronal differentiation.** CRISPRi-i[3]N iPSCs were differentiated using doxycycline-induced expression of NGN2 based on previously described methods [70,99]. iPSCs were collected using Accutase (GIBCO/Thermo Fisher Scientific, cat. no. A1110501) by aspirating medium, washing 1x with DPBS, and then incubating the cells in Accutase at 37C for 10min. Accutase was then diluted with mTesR1 medium supplemented with 10nM Y-27632 dihydrochloride ROCK inhibitor (Tocris; Cat. No. 125410), and the iPSCs pelleted and then resuspended in fresh mTeSR1 with RI for counting. iPSCs were then plated in Differentiation Medium, comprised of Neurobasal Plus Medium (GIBCO/Thermo Fisher Scientific, cat. no. A3582901), 1x N2 supplement (GIBCO/Thermo Fisher Scientific, cat. no. A1370701), 1x B27 Plus supplement (GIBCO/Thermo Fisher Scientific, cat. no. A3582801), 1X MEM Non-Essential Amino Acids (GIBCO/Thermo Fisher Scientific; Cat. No. 11140–050), and 1X GlutaMAX Supplement (GIBCO/Thermo Fisher Scientific; Cat. No. 35050–061), supplemented with 10nM ROCK inhibitor, and 2 mg/mL doxycycline hydrochloride (Sigma-Aldrich; Cat. No. D3072) to induce expression of mNGN2. iPSCs were plated at a

concentration of 5-7x$10^5$ cells per well on 6 well plates coated with Matrigel, Growth Factor Reduced (GFR) Basement Membrane Matrix, LDEV-free (Corning; Cat. No. 354230), diluted to 0.5mg/plate. After three days, pre-differentiated cells were collected as above and resuspended for counting and replating in Differentiation Medium supplemented with 10nM ROCK inhibitor, 2 mg/mL doxycycline hydrochloride, 1x CultureONE supplement (GIBCO/Thermo Fisher Scientific, cat. no. A3320201), 10ng/mL NT-3 (PeproTech; Cat. No. 450–03), 10ng/mL BDNF (PeproTech; Cat. No. 450–02), 10ng/mL GDNF (PeproTech; Cat. No. 450–10), and 200µM L-ascorbic acid (Sigma-Aldrich, cat. no. A8960). 6-well plates or 24-well glass-bottomed dishes (Cellvis, Cat. No. P24-1.5H-N) were prepared for pre-differentiated cells by coating with 100 ug/mL poly-L-ornithine (Sigma-Aldrich, Cat. No. P3655) overnight at 37C, washing 3x with H2O, and drying overnight at room temperature. The plates were then pre-incubated with plain Neurobasal Plus media at 37C while pre-differentiated cells were prepared for replating. Pre-incubation media was then aspirated, and pre-differentiated cells plated at a density of 5x$10^5$ cells/well for 6-well plates for biochemistry assays and at 5x$10^4$ cells/well on 24-well plates for imaging. After cells attached (about 1hr), an additional volume of media (2mL per 6 well and 0.5mL per 24 well), prepared as above for replating, was added, but without ROCK inhibitor and with the addition of laminin at 2mg/mL (final concentration per well of 1mg/mL). Thereafter, half media changes were performed 1-2x per week with Differentiation Medium supplemented as above for replating with laminin at 1mg/mL and without doxycycline and ROCK inhibitor.

**Lentiviral transduction of iPSCs.**   Lentivirus was added dropwise onto iPSCs growing in mTesR1 medium supplemented with 10nM ROCK inhibitor plated at 5x$10^4$ cells per well on 6-well plates coated with hESC Matrigel. 48 hours after transduction, 1ug/mL puromycin was added to transduced cells and to a non-transduced well. After 2 days, once all cells in the non-transduced control well were dead, the iPSCs were taken off ROCK inhibitor. Puromycin treatment was maintained through the first passage post-transduction.

## Molecular cloning

**TDP-43 Aggregation reporter cloning.**   Doxycycline-inducible TDP-43 ΔNLS mammalian expression vector was constructed based on previous work [18]. Human wild-type TDP-43 was amplified in two separate PCR reactions to exclude the wild type NLS. TDP-43 amplicons were reassembeled using Gibson cloning (NEB #E2611), where the Gibson overlap contained mutated NLS. The assembled TDP-43ΔNLS was then amplified to add Golden Gate cloning sites on either side of the gene, and then inserted via Golden Gate cloning downstream of a doxycycline inducible TRE promotor and mClover3, producing TRE-mClover3-linker-TDP-43ΔNLS.

**Individual sgRNAs for CRISPR KO and KD.**   sgRNA sequences for targeted screen validation were taken from the Brunello library and forward and reverse sgRNA sequences were ordered as primers. sgRNAs primers were phosphorylated and annealed before being inserted by Golden Gate cloning into BsmBI cloning sites. CRISPR KO sgRNAs were cloned into lenti-Guide-Puro (Addgene #52963). For CRISPRi sgRNAs were designed using CRISPick (Broad Institute) and cloned into dCas9-KRAB (Addgene #71236).

**SRRD-mRuby3.**   Full length wild type SRRD expression vectors were amplified from cDNA (Genscript, Clone ID OHu31013) and inserted using Gibson Assembly downstream of a doxycycline inducible TRE promoter and upstream of mRuby3. Codon optimized geneblocks of SRRD truncations flanked with Golden Gate cloning adapters (IDT Technologies) were inserted via Golden Gate cloning downstream of a doxycycline inducible TRE promoter and upstream of mRuby3.

**SRRD-V5-APEX2.**   SRRD and APEX2 were amplified cloned via Gibson Assembly into pLEX_307 (Addgene #41392) downstream of an EF1a promoter. SRRD was amplified from the SRRD-mRuby3 vector described above to contain 5' overlap to pLEX_307, a 3' V5 tag, and 3' Gibson overlap to APEX2. APEX2 was amplified from pcDNA5-FRT-TP-APEX2-GFP (Addgene # 129640) to contain 5' Gibson overlap to SRRD-V5 and 3' Gibson overlap to pLEX_307. pLEX_307 was digested with NheI-HF and Mlul-HF before SRRD and APEX2 amplicons were inserted via Gibson Assembly.

## FACS Pulse-Shape Analysis detection of aggregates

Transfected cells were lifted and passed through a 35μm cell strainer (Falcon, 352235). Cells were gated to have a narrow range of FCS and SSC values. Autofluorescence was detected by the 405nm laser and 450/50 filter. TDP-43 reporter fluorescence was detected using the 488nm laser and 515/510 filter, and aggregation was quantified by comparing the height (FITC-H) to the width (FITC-W) of the fluorescence channel.

## Cell engineering

**Polyclonal sgRNA KO and KD.**   Low passage 293Ts were transduced with sgRNA lentivirus targeting individual gene loci or AAV5 targeting control. At least 2 individual sgRNAs taken from the Brunello library were used to validate gene KO effect on TDP-43 aggregation. Four CRISPRi sgRNAs targeting SRRD and 2 AAV5 targeting control sgRNAs were used as an orthogonal test of SRRD depletion. Cells were transduced by mixing lentivirus, polybrene infection reagent (Sigma #TR1003G, 1:1000), with cells while in suspension, then plating on 0.1% gelatin coated wells. 24hrs after lentiviral infection cells infected with sgRNAs were selected using puromycin (1ug/ml, ThermoFisher #A1113803). Two timepoints of protein depletion were tested on TDP-43 aggregation. At one week or two weeks post puromycin treatment, KO cells were transfected with TRE-TDP-43 ΔNLS, and 24hrs after doxycycline induction cells were analyzed by FACS PulSA for changes in TDP-43 aggregation.

**Clonal SRRD KO generation and characterization.**   Polyclonal SRRD CRISPR KO 293Ts (Synthego) were single cell sorted into 96-well plates. After expansion, genomic DNA was extracted from each clone and the area around the sgRNA target was sequenced to confirm Cas9 cutting by the presence of indels. Sequences were manually searched for premature stop codons, as well as analyzed using Synthego's Interference of CRISPR Edits (ICE) tool to predict gene KO (Hsiau, et al, (2018). bioRxiv). Clonal SRRD KO cells were also transfected with the TDP-43 aggregation reporter and analyzed by FACS PulSA.

**SRRD-mRuby3.**   Low passage 293Ts were transduced at low MOI with TRE-mRuby3, TRE-SRRD-mRuby3, TRE-SRRD_NTD-mRuby3, or TRE-SRRD_no_NTD-mRuby3 lentivirus and polybrene infection reagent (Sigma #TR1003G, 1:1000). Successfully infected cells were isolated by sorting and collecting mRuby3 positive cells into 1 well of a 6-well plate per line and expanded.

**SRRD-V5-APEX2 stable and clonal cell line generation and characterization.**   Low passage 293Ts were transduced at low MOI with EF1a-SRRD-V5-APEX2 lentivirus and polybrene infection reagent (Sigma #TR1003G, 1:1000). Successfully infected cells were selected using puromycin (1ug/ml, ThermoFisher #A1113803) for at least 3 days. To ensure transgene expression cells were maintained on puromycin. Expression and localization of APEX2 tagged SRRD was assessed by both immunofluorescence and western blot against V5. To generate clonal cell lines the polyclonal stable line expressing SRRD-V5-APEX2 was single cell sorted into 96-well plates and expanded. Western blot against V5 was used to find clones lowly

expressing SRRD-V5-APEX2. Immunofluorescence was also used to confirm tags did not alter SRRD localization.

## Genome-wide screen in 293T

**Genome-wide plasmid library preparation.** Brunello genome-wide sgRNA library containing an average 4 sgRNAs per gene and 1000 non-targeting control sgRNAs was purchased from Addgene (#73178). The library was transformed into electrocompetent cells (Lucigen #60242–1) and recovered at 32C for 16-18hrs for prevent recombination. Plasmid DNA was sequenced to confirm library distribution and sgRNA representation.

**Lentivirus titering in 293Ts.** To ensure low MOI lentiviral transduction, library sgRNA lentivirus was titered by plating 2x10^6 293Ts per well of a 12-well plate and different volumes of virus were mixed while the cells were in suspension along with polybrene infection reagent (Sigma #TR1003G, 1:1000). Plates were spinfected by centrifugation at 1000xg, 1hr, 37C. After ~16hrs each well was split into duplicate wells, one receiving no treatment and the other treated with puromycin (1ug/ml, ThermoFisher #A1113803). After three days, cells from each well were lifted and counted, and the ratio of live cells in the +/- puromycin wells was calculated. The virus volume that achieved approximately 30% of cell survival after puromycin treatment was used for the genome-wide screen.

**FACS-based CRISPR knockout screen for TDP-43 aggregation.** Low passage 293Ts grown to ~85% confluency before being lifted and counted. To achieve >1000x coverage, 288x10^6 cells were mixed with polybrene infection reagent (Sigma #TR1003G, 1:1000) and the Brunello genome-wide sgRNA library at low MOI (titer calculated above). After thorough mixing, 2x10^6 cells were plated per well in 12-well plates, and then spinfected by centrifugation at 1000xg, 1hr, 37C. Following overnight incubation, all cells were lifted from 12w plates, counted, and plated in 15cm plates at 7x10^6 cells per plate. Puromycin (1ug/ml, ThermoFisher #A1113803) was added to plates to select for transduced cells. Cells were split every 3–4 days over the next 14 days and maintained on puromycin selection. At each split, cells were counted and 160x10^6 cells were replated across 20x 15cm plates. All remaining cells were discarded.

On day 14 of puromycin selection Brunello sgRNA library transduced cells were lifted and counted. Library cells were plated across 15cm plates to be 75% confluent 24hrs post plating. All remaining cells were frozen and stored in liquid nitrogen. 24hrs after plating, cells were transfected with 20ug of TDP-43 aggregation reporter and 136ul of poluethylenimine HCL MAX transfection reagent per plate. Media was changed 17hrs later to contain 1ug/ml doxycycline hyclate.

Two days post transfection cells were lifted and fixed in 2% paraformaldehyde. Cells were applied to 35μm filter (Falcon, 352235) before FACS analysis and collection. Cells were gated to have a narrow range of FCS and SSC values to select for live, single cells. Autofluorescence was detected by the 405nm laser and 450/50 filter. TDP-43 reporter fluorescence was detected using the 488nm laser and 515/510 filter, and aggregation was quantified by comparing the height (FITC-H) to the width (FITC-W) of the fluorescence channel. Narrow gates were defined around the aggregate and non-aggregate populations such that approximately 5% of cells from each population were collected. A total of 57x10^6 and 54x10^6 aggregate and non-aggregte cells were collected, respectively. Collected aggregate and non-aggregate cells were then divided into two samples each to provide technical replicates for gDNA extraction, PCR, and sequencing. Sorted cells were then pelleted and stored at -20C until DNA extraction.

**DNA Extraction, PCR amplification and next generation sequencing.** Cell pellets were thawed on ice then resuspended in lysis buffer (50mM Tris, 50mM EDTA, 1% SDS, pH 8).

After resuspension proteinase K (Qiagen #19131) was added to each sample and then incubated at 55C overnight. After overnight incubation RNase A (Qiagen #19101, 10mg/ml) was added to each sample, mixed thoroughly, then incubated for 30min at 37C. Samples were immediately placed on ice after incubation with RNaseA, where pre-chilled 7.5M ammonium acetate was added to cooled samples to precipitate proteins. The samples were then vortexed and spun at >4,000xg for at least 10min. Centrifugation step was repeated when it was observed that all proteins did not pellet. The supernatant was then transferred to fresh tubes, where 100% isopropanol was added to precipitate the genomic DNA. Next, 70% ethanol was added to further purify the genomic DNA, and samples were spun at top speed to pellet the DNA. Finally, genomic DNA was air dried before resuspension in TE buffer before being quantified by Nanodrop.

sgRNA sequences were PCR amplified with custom primers targeting the genome-integrated sgRNA backbone and containing Illumina adapters and unique barcodes for each sample to allow for multiplexing. PCR products were gel extracted and quantified by Qubit dsDNA HS assay (ThermoFisher Scientific# Q32851). All samples were then pooled in equimolar ratios and sequenced using Ilumina NextSeq 500/500 v2 75 cycle kit (Illumina #20024906). Amplifications were carried out with 1x8 cycles for sample index reads and 1x63 cycles for the sgRNA.

**Screen data analysis.** Raw fastq files were trimmed to remove sequences that flank the 20bp and mapped to the sgRNA library using Bowtie. sgRNA counts were then loaded to R and the following steps were performed to calculate a phenotype and p-value for each gene. Counts were first normlized by read depth by dividing read count by sample mean, mutiplying by a million and adding 1 pseudocount. Next, for each sgRNA, we calculate the fold change between the aggregation positive to aggreation negative sample. Fold changes are corrected for increased variance at low mean values by computing a local Z score, which is calculated by ranking all the sgRNAs by mean value between the two conditions and calculating a Z score using the 2000 sgRNA window around each sgRNA. These local Z scores are then used to calculate a *phenotype* and *p-value* for each gene as follows: *phenotype* as the mean of the two sgRNA with the maximum absolute local Z score. *P-value* is calculated by taking the mean of all sgRNAs against a gene and comparing to an empirical distribution of mean local Z-score generated by 100,000 permutations of gene to sgRNA associations.

## Immunohistochemistry

293T cells were plated on coverslips coated with 0.1% gelatin and fixed with 4% paraformaldehyde at room temperature for 20min, then permeabilized with DPBS, 1% BSA, and 0.25% Triton X-100 at room temperature for 30min. Primary antibody in PBS with 1% BSA and 0.25% Triton X-100 was applied and for 1hr at room temperature unless otherwise noted. Following three washes with 1x PBS secondary antibody was applied in PBS with 1% BSA for 1hr at room temperature. Following three washes with 1x PBS coverslips were mounted onto slides using ProLong Glass Antifade Mountant with NucBlue Stain (ThermoFisher #P36981). Images were acquired using a Leica SP8 confocal using the 63x oil immersion objective. Primary antibodies used for immunohistochemistry are found in S8 Table.

## Proximity ligation assay

293T cells were plated on coverslips coated with 0.1% gelatin and fixed with 4% paraformaldehyde at room temperature for 20min, then permeabilized with DPBS, 1% BSA, and 0.25% Triton X-100 at room temperature for 30min. Additional blocking, antibody incubation, probe incubation, ligation, and rolling circle amplification were carried out using Duolink In Situ

Red Starter Kit Mouse/Rabbit (Sigma DUO92102) and following manufacturer recommendations. Assay was adapted for use in 24 well glass bottom plate by adding DAPI stain to final wash rather than mounting on glass slide. Images were acquired in the TexasRed channel using Nikon Ti-2 Eclipse epifluorescent scope.

## Imaging analysis

Processing of image files, including LUTs adjustment and cropping, was done in FIJI. Aggresome quantification was performed manually by counting the number of perinuclear HDAC6 and VIM positive bodies and dividing by the total number of cells. VIM encircled TDP-43 quantification was done manually by counting the number of TDP-43 perinuclear foci with at least partial VIM staining around the periphery, divided by the total number of TDP-43 foci. Colocalization analysis was performed by cropping images to analyze one cell at a time, and JACoP FIJI plugin was used to calculate Pearson's correlation coefficient.

## Western blotting

For western blot analysis, a minimum of one well of a 6-well plate at 75% confluence was lifted and pelleted before lysis in 1x RIPA buffer supplemented with 1x Protease Inhibitor Cocktail (Sigma #P8340). Lysed cells were then spun at 14000 rpm for 10min at 4C to isolate and discard DNA pellet. Next, 1x Laemmli sample buffer and 0.05% beta-mercaptoethanol were mixed with equal volume of cell lysis before boiling for 10min at 100C. Denatured protein was loaded into 4–15% mini-protean gels (BioRad #4561084) before transfer to nitrocellulose membrane using standard wet transfer protocol. The membranes were then blocked for 1hr at room temperature with either 1x Tris buffered saline with 1% casein, 1x PBS with 5% BSA, or 1x Tris buffered saline with 5% milk. Membranes were incubated with primary antibody overnight at 4C, washed 3x with Tris buffered saline with 0.1% Tween-20 (BioRad #1706435), then incubated with secondary antibody for 1hr at room temperature. Membranes were washed 3x with Tris buffered saline with 0.1% Tween-20 before imaging using the Odessey Fc imaging system (LI-COR) and processing using Image Studio software. Densitometry was performed using FIJI. Primary antibodies used for western blot can be found in S8 Table.

## APEX2 proximity labeling

**Biotin-phenol cell labeling.** SRRD-V5-APEX2 clonal cells were plated across 10cm plates coated with fibronectin (Sigma, #fc010, 1:100 in DPBS). Cells were plated so they were 80% confluent the day of cell labeling. The day of, cells were incubated with 0.5μM biotin-phenol (BP; Iris-Biotech) dissolved in DMSO for 30 minutes. After BP incubation, targeted wells were treated with 30% hydrogen peroxide ($H_2O_2$; Sigma-Aldrich) diluted with distilled water making a 1mM solution for 1-minute, excluding control wells. Wells were gently agitated. BP solution containing $H_2O_2$ was poured out and immediately replaced with a quenching solution. Labeling reaction is quenched with quenching solution consisting of DPBS, 10mM sodium azide (Sigma-Aldrich) dissolved with distilled water, 10mM sodium ascorbate (Sigma-Aldrich) dissolved in distilled water and 5mM Trolox (Sigma-Aldrich) dissolved in DMSO. Both control and targeted wells were quenched three times with quencher solution with gentle agitation so as to not detach any cells.

**Streptavidin-biotin pull down.** All samples, lysis and wash buffers were kept on ice throughout the procedure. For a 10cm plate, 885ul of 1x RIPA lysis buffer supplemented with 1x protease inhibitor cocktail (PIC; Sigma-Aldrich) was added. Lysed cells were then spun at 14000 rpm for 10min at 4C to isolate and discard DNA pellet. Approximately 20% of the

whole cell lysate was saved for 'input' analysis. For each sample, 60ul aliquots of Pierce strepta-vidin magnetic beads (Thermo Scientific; Lot: WB319828) were washed twice with 2mL of RIPA lysis buffer supplemented with 1x protein inhibitor cocktail (PIC; Sigma-Aldrich), 1mM PMSF (G-Biosciences), and quenchers (10mM sodium azide, 10mM sodium ascorbate and 5mM Trolox). Magnetic bead rack (VWR) was used to collect the beads in order to remove the washes. 590ul of whole cell lysate sample was incubated with the washed streptavidin magnetic beads for 1 hour at room temperature on a HulaMixer sample mixer (Invitrogen). To stimulate rotation, an additional 1mL of RIPA lysis buffer was added. The beads were pelleted using DynaMag magnetic rack (Invitrogen) to collect and save the supernatant. The saved superna-tant is designated the 'flow through'. Each bead sample was washed in the following series of buffer solutions at 2mL each to remove non-specific binders: twice with RIPA lysis buffer, once with 1M KCl, once with 0.1M Na2CO3, once with 2 M urea in 10 mM Tris-HCl, pH 8.0, and twice with RIPA lysis buffer. The beads were washed three times in DPBS to remove traces of wash buffers. Beads were snap-frozen and placed in -80˚C until samples were sent out for mass spectrometry analysis.

**Analysis.**   Missing values for MG132 and non-stressed groups were imputed separately using missForest before filtering to exclude lowly expressed proteins. Differential expression analysis using limma and voom, and gene set enrichment analysis using CORUM database of protein complexes was performed on ranked list of hits. Top 100 ranked proteins were used for STRING analysis.

## Quantitative proteomics

**Sample preparation.**   Five replicates of wild type and two SRRD clonal KO lines were plated on gelatin coated 10cm plates. When they were ~85% confluent 24hrs after plating, cells were lifted and pelleted at 500 rcf for 3min at room temperature. Pellets were washed 3x with ice cold PBS then snap frozen and stored at -80C until sample submission.

See below for mass spectrometry acquisition.

## Analysis

Missing values for WT and SRRD KO samples were imputed separately by generating distribu-tions corresponding to the number of missing counts and sample from these distributions to replace missing counts. Differential expression analysis was performed using limma and voom. Gene set enrichment analysis was performed on a ranked list of hits, and top 100 ranked proteins were used for STRING analysis.

## RNAseq

**Sample preparation, submission, and analysis.**   Three replicates of wild type and two SRRD clonal KO lines were plated on gelatin coated 10cm plates. When they were ~85% con-fluent 24hrs after plating, cells were lifted and pelleted at 500 rcf for 3min at room temperature. Pellets were washed 3x with ice cold PBS then snap frozen and stored at -80. RNA extraction was performed using RNeasy mini kit (Qiagen) following manufacturer's directions. Extracted RNA samples were shipped to Genewiz, where RNAseq using PolyA selection for eukaryotic mRNA species with 5–10 million reads per sample was performed. Raw FASTQ files were mapped to the genome using Salmon. Differential expression analysis was performed using limma and edgeR in R.

## Mass spectrometry acquisition for quantitative proteomics and APEX2 proximity labeling

**In solution digestion.** Samples were solubilized and digested with the iST kit (PreOmics GmbH, Martinsried, Germany) per manufacturers protocol. Briefly, the resulting pellet was solubilized, reduced, and alkylated by addition of SDC buffer containing TCEP and 2-chloroacetamide and heated to 95˚C for 10 minutes. Proteins were enzymatically hydrolyzed for 1.5 hours at 37˚C by addition of LysC and trypsin. The resulting Peptides were de-salted, dried by vacuum centrifugation, and reconstituted in 0.1% TFA containing iRT peptides (Biognosys).

**Mass spectrometry: Data Dependent Acquisition (DDA for APEX proximity labeling).** Samples were randomized and 2ug of each was analyzed on an Exploris 480 mass spectrometer (Thermofisher Scientific San Jose, CA) coupled with an Ultimate 3000 nano UPLC system and an EasySpray source. Peptides were loaded onto an Acclaim PepMap 100 75μm x 2cm trap column (Thermo) at 5uL/min, and separated by reverse phase (RP)-HPLC on a nanocapillary column, 75 μm id × 50cm 2μm PepMap RSLC C18 column (Thermo). Mobile phase A consisted of 0.1% formic acid and mobile phase B of 0.1% formic acid/acetonitrile. Peptides were eluted into the mass spectrometer at 300 nL/min with each RP-LC run comprising a 90 minute gradient from 3% B to 38% B. For Data Dependent Acquisition (DDA), the mass spectrometer was set with a master scan at 120000 MS resolution, a scan range of 300–1400, AGC target set to standard, maximum injection time set to auto, and dynamic exclusion set to 30 seconds. Charge state 2–5 were included and Top 15 data dependent MS2 scans collected at 45000 resolution with normalized AGC target at 300%. Maximum injection time and HCD NCE were set to auto and 30, respectively.

**Mass spectrometry: Data Independent Acquisition (DIA for quantitative proteomics).** Samples were randomized and 2ug of each was analyzed on a QExactive HF mass spectrometer (Thermofisher Scientific San Jose, CA) coupled with an Ultimate 3000 nano UPLC system and an EasySpray source. The LC settings were the same as what we described in the previous section. Data was acquired using Data Independent Acquisition (DIA). Mass spectrometer settings were: one full MS scan at 120,000 resolution and a scan range of 300–1650 m/z with an AGC target of 3e6 and a maximum inject time of 60ms. This was followed by 22 (DIA) isolation windows with varying sizes at 30,000 resolution. AGC target and injection time were set to 3e6 and auto, respectively. The default charge state was 4, the first mass was fixed at 200 m/z and the normalized collision energy (NCE) for each window was stepped at 25.5, 27 and 30.

**System suitability and quality control.** The suitability of Exploris 480/QE HF mass spectrometers was monitored using QuiC software (Biognosys, Schlieren, Switzerland) for the analysis of the spiked-in iRT peptides. Meanwhile, as a measure for quality control, we injected standard E. coli protein digest in between samples (one injection after every 4 biological samples) and collected the data in the Data Dependent Acquisition (DDA) mode. The collected DDA data were analyzed in MaxQuant [100] and the output was subsequently visualized using the PTXQC [101] package to track the quality of the instrumentation.

**Database searching for DDA raw files.** MS/MS raw files were searched against reference human protein sequence database including reviewed isoforms from the Uniprot using MaxQuant version 1.6.14.0. Carbamidomethyl of Cys was defined as a fixed modification. Oxidation of Met and Acetylation of protein N-terminal were set as variable modifications. Trypsin/P was selected as the digestion enzyme, and a maximum of 3 labeled amino acids and 2 missed cleavages per peptide were allowed. The false discovery rate for peptides and proteins were set at 1%. Fragment ion tolerance was set to 0.5 Da. The MS/MS tolerance was set at 20 ppm. The

minimum peptide length was set at 7 amino acids. The rest of the parameters were kept as default.

**Database searching for DIA raw files.** Protein identification/quantification was performed in Spectronaut [102] version 15. Whole proteome analysis was conducted in the DirectDIA mode using reference human protein sequence database including reviewed isoforms. Carbamidomethyl of Cys was defined as a fixed modification. Oxidation of Met and Acetylation of protein N-terminal were set as variable modifications. The cutoff values for precursor Qvalue, precursor PEP, and protein Qvalue were set at 0.01, 0.2, and 0.01, respectively. The quantitation was performed at MS2 level and the intensity values were normalized using default cross-run normalization algorithm. The rest of the parameters in Spectronaut were kept as default.

## Yeast growth assays

**Yeast strain construction.** To generate the yeast strains for this study, strain BY4741 was transformed with plasmid pEB413GAL expressing TDP-43, FUS, or alpha-synuclein using standard PEG/lithium acetate transformation procedure [103]. Transformants were selected for with SD-HIS dropout media. SRRD, related proteins, and an empty vector control were expressed with plasmid pEB416GAL. After transformation, yeast colonies harboring pEB413-GAL and pEB416GAL plasmids were selected for by SD-HIS/URA dropout media.

**Yeast liquid growth assay.** Yeast strains harboring both pEB413GAL and pEB416GAL plasmids were inoculated in a raffinose -HIS/URA pre-induction culture and grown to saturation for 16 hours at 30°C. For gene induction and growth measurement, yeast strains were inoculated to galactose -HIS/URA media at initial $OD_{600}$ of 0.02 in 300 μL and grown with shaking at 30°C for 48-hours in a Bioscreen plate reader. The $OD_{600}$ was measured every 15 minutes. All growth assays were performed in biological triplicate.

**Yeast agar spotting assay.** Yeast strains were grown in pre-induction culture, in raffinose -HIS/URA, for 16 hours at 30°C. These cultures were then normalized to $OD_{600}$ of 2.0 and each sample was diluted in a five-fold dilution series across a 96-well plate. A 96-pin replicator tool was used to transfer yeast from the 96-well liquid plate to SD -HIS/URA and galactose -HIS/URA agar plates. Yeast spots were grown at 30°C for 48 hours and imaged. All spotting assays were performed in biological triplicate.

**Imaging of TDP-43 aggregates in yeast.** Plasmid pEB413GAL-TDP43-YFP was used to visualize TDP-43 aggregates. Yeast strains harboring pEB413GAL-TDP43-YFP and pEB416-GAL plasmids were grown in pre-induction raffinose -HIS/URA media for 16 hours at 30°C. These strains were inoculated to galactose -HIS/URA media at initial $OD_{600}$ of 0.6 in 6mL and grown with shaking at 30°C for 5 hours before imaging. Yeast strains were imaged using Evos M5000 microscope. 100 cells were imaged for each strain and the number of TDP-43-YFP foci per cell were manually counted for comparison.

## Supporting information

**S1 Text. Figs S1-S15.**
(PDF)

**S1 Table. TDP-43 screen hits (more aggregation).**
(XLSX)

**S2 Table. TDP-43 screen hits (less aggregation).**
(XLSX)

**S3 Table. APEX no stress data.**
(XLSX)

**S4 Table. Processed quantitative proteomics data.**
(XLSX)

**S5 Table. APEX data with MG132.**
(XLSX)

**S6 Table. RNAseq data.**
(XLSX)

**S7 Table. Spacer sequences for sgRNAs used in validation studies.**
(XLSX)

**S8 Table. Antibodies used.**
(XLSX)

**S9 Table. Plasmids submitted to AddGene.**
(XLSX)

## Author Contributions

**Conceptualization:** Katelyn M. Sweeney, Ophir Shalem.

**Data curation:** Katelyn M. Sweeney, Ophir Shalem.

**Formal analysis:** Katelyn M. Sweeney, Ophir Shalem.

**Investigation:** Katelyn M. Sweeney, Sapanna Chantarawong, Edward M. Barbieri, Greg Cajka, Matthew Liu, Hossein Fazelinia, Bede Portz, Katie Copley, Tomer Lapidot, Lauren Duhamel, Phoebe Greenwald, Naseeb Saida.

**Methodology:** Sapanna Chantarawong.

**Project administration:** Ophir Shalem.

**Supervision:** Lynn Spruce, Reut Shalgi, James Shorter, Ophir Shalem.

**Validation:** Katelyn M. Sweeney.

**Visualization:** Katelyn M. Sweeney.

**Writing – original draft:** Katelyn M. Sweeney.

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
