## [Decision Letter · Decision Letter 0]

22 Aug 2023

Dear Dr Shalem,

Thank you very much for submitting your Research Article entitled 'CRISPR screen for protein inclusion formation uncovers a role for SRRD in the regulation of intermediate filament dynamics and aggresome assembly' to PLOS Genetics.

The manuscript was fully evaluated at the editorial level and by three independent peer reviewers. As you will see, the reviewers are interested in and enthusiastic about the work; however, there are a number of concerns that will need to be addressed before moving forward, and that will likely require additional data. 

We therefore ask you to modify the manuscript according to the review recommendations. Your revisions should address the specific points made by each reviewer.

We hope to receive your revised manuscript within the next 60 days. If you anticipate any delay in its return, we would ask you to let us know the expected resubmission date by email to plosgenetics@plos.org.

Yours sincerely,

Gregory S. Barsh

Editor-in-Chief

PLOS Genetics

Gregory Copenhaver

Editor-in-Chief

PLOS Genetics

Reviewer's Responses to Questions

**Comments to the Authors:**

Reviewer #1: In the respective manuscript title, “CRISPR screen for protein inclusion formation uncovers a role for SRRD in the regulation of intermediate filament dynamics and aggresome assembly”, Sweeney et al. develop and perform a novel FACS-based CRISPR screen to identify modifiers TDP-43 aggregation in HEK293 human cells. In addition to finding established regulators of proteostasis, they uncover and characterize SRRD as top modifier of TDP-43. Moreover, they find that this poorly characterized protein plays important roles both in intermediate filament stability and aggresome formation. Structure function analysis highlights the importance of the N-terminal low complexity domain on SRRD. The findings from this manuscript are interesting and provide a novel modifier of TDP-43 in SRRD. Furthermore, the research introduces a unique screening strategy, which could be used for other aggregates or macromolecular assemblies within the cell. Overall, this study highlights the links between subcellular architecture and disease-protein assembly dynamics which is highly significant and continues to emerge as critical factor in disease pathogenesis. The manuscript and screening hits will be a valuable resource for these authors and other labs to build upon for years to come. Thus, this manuscript is suitable for publication in PLOS genetics. I have included some suggests to improve the manuscript below.

In the manuscript, the authors do a nice job providing the precise methods of analysis but it does bloat the main text, making it hard for the reader get through the paper. Is it possible for these methods to be included in the figure legends or in the methods section? Moreover, most of the fluorescence micrographs throughout the paper are small and should either be enlarged or zoom panels included for ease of reading.

Figure 1

Included is a nice control showing the impact of the mClover tag on TDP-43 aggregation. Moreover, identification of several chaperone proteins provides good validation for their screen.

Screens and much of the validation throughout the paper were done in HEK293 cells. While neuronal lines were used in a small portion of the paper, the authors should try to include more validation within neurons throughout.

How does SRRD KO impact aggregation of other metastable, disease-linked protein?

Figure 2

Characterization of SRRD was largely performed in HEK293 cells. Perhaps including some characterization of SRRD in human neurons.

CRISPRi is good alternate validation of SRRD as a pro-aggregation factor.

Does the use of H2O2 for Apex labeling impact SRRD interactions with keratin (which form heterodimers via disulfide linkages in oxidizing environment)? Perhaps even just show colocalization of SRRD with keratin labeled intermediate filaments?

Figure 3

3A- their point about VIM in SRRD KO cells appearing fragmented and disorganized. It’s not clear by the images presented. Since this is important for their overall interpretations about the IF network being disrupted, its recommended to provide clearer pictures or some complementary biochemistry that network is indeed disrupted.

How does expression of SRRD rescue compare to endogenous SRRD levels? Maybe qPCR to examine.

Is the reduction in INA and MAP2 signal in axonal and dendritic projections of SRRD KO cells proportion to their reduced signal in the cell body? The thought being that these results could simply be due to overall reduced steady-state levels in the cell. Can the investigators ratio the INA and MAP2 fluorescence signal in the dendrites/axons with that of the cell body to see if there is a difference between wild type and SRRD KO cells.

Is it known whether disrupting intermediate filament formation is sufficient to impact TDP-43 aggregation? If so, it would be worth including the citation to complement SRRD data and bolster the involvement of IF network.

Figure 4

In Fig 4c, the authors state that SRRD KO cells have more diffuse TDP-43 with lack of VIM cages. The authors should include zoom panel to better show this point.

The inclusion of SQSTM1 in this figure might confuse reader. The story begins with a screen that identifies SRRD, which moves to IF dynamics and next aggresome formation. Now including autophagosome formation might further complicate the interpretations and dilute the impact of the other observations.

Figure 5

Fig 5e, please included zoom panel to better highlight the colocalization.

Figure 6

Is there a reason why toxicity or growth studies were not conducted in human cells or neurons? Its not essential but this should be mentioned at the beginning of the paragraph in this section to better explain why yeast models were employed.

Reviewer #2: In this paper Sweeney and co-authors set out to identify candidate mechanisms able to modulate aggregation of TDP-43 - a partially disordered RNA-binding protein which is mutated in ALS and aggregates in many other neurodegenerative conditions. I think the experiments in this study are robust and that overall this work succeeds in identifying SRDD as a new important player in aggresome formation. They also discover that SRRD likely acts via interactions with IF proteins and acts more generally in proteotoxic stress, providing a nice example of a connection between IF spatial regulation and cellular proteostasis. I also think the paper nicely shows the power of high-throughput genetic approaches to gather mechanistic insights on protein (dys)regulation. However, I think the paper can improve in clarity - especially in the discussion. In the current form of the text, it is unclear whether the authors believe their data supports a picture in which organised TDP-43 aggregation can be beneficial for the cell or rather is detrimental. I think - on the basis of their data - they can risk a more blunt take on this, or at least more clearly argue if they think SRRD should be up-regulated or down-regulated in human cells to counter-act protein-induced toxicity.

These are the points I would like the authors to address:

⁃ Can they explicitly show that expression of delta-NLS-TDP in HEK293T cells does not lead to impairment of cell viability in the time frame of their experiments?

⁃ If that is the case, why do they think no toxicity is seen in this cell type while a clear effect is seen when TDP-43 is expressed in yeast?

⁃ Would this change if they expressed a fALS mutant?

⁃ Given the results I think this sentence is misleading: “Taken together, the yeast toxicity and mammalian cell data suggest that SRRD acts through its NTD to reduce both toxicity and aggregation of unfolded proteins in orthogonal model systems.“ 1)Knocking out SRDD leads to a decrease in number of foci in HEK. 2) Expressing SRRD in yeast does not alter the number of foci. Thus I think that stating that SRRD “reduces” aggregation of unfolded protein is incorrect. What I gather from the experiments is instead that SRDD is required for IF regulation and “active” organised protein aggregation. I think this message and its implications should be stated way more clearly.

⁃ On this line I think the discussion should also put the results into context when it comes to TDP-43 induced toxicity. Several lines of work now suggests that efficient compartmentalisation of the TDP-43 in aggregates can actually relieve protein induced toxicity. I believe the results presented here should also feed into that discussion, while adding novel mechanistic insights.

⁃ Expression of the SRRD-NTD alone partially rescues TDP-43 induced toxicity in yeast, without altering the number of foci. On the other hand it alters the number of inclusions in HEK293T where - as far as this paper shows - TDP-43 expression is not toxic. Can this difference open the way to understand the causes of protein-induced toxicity and its relation to aggregation?

Minor:

⁃ This needs rephrasing “An alternative hypothesis could be that SRRD is acting as a type of chaperone, both assisting in the formation of IF networks which are composed of fibrillar proteins that are prone to aggregation(Didonna and Opal 2019), and interacts with unfolded proteins in aggrosomes in a similar manner.“

⁃ At several points in the text they refer to aggresomes as “aggrosomes”. Probably a typo, but needs to be fixed.

⁃ I suggest to deposit the fastQ raw sequencing reads + processed files on GEO

Reviewer #3: Summary

The manuscript by Katelyn M Sweeney et al. presents a study of regulators of proteostasis by directly examining the protein inclusion formation instead of targeting downstream proteinopathy-associated cellular toxicity. The authors performed a genome-wide CRISPR-Cas9 KO screen. They utilized the PulSA method to quantify fluorescent TDP-43 aggregation at a single-cell level and identified a specific subset of proteins that affected inclusion formation. The authors applied APEX2 proximity proteomics for SRRD, one of the top hits from their screen and also employed imaging in cellular systems with induced aggresome formation to elucidate its functional significance. They try to show that the loss of SRRD affects the organization of intermediate filaments like Vimentin (VIM) and that it localizes to aggresomes and is a necessary component for the assembly of aggresomes. They further turned to exogenous expression of aggregation-prone proteins in yeast and found that SRRD may reduce aggregate-induced toxicity. Overall, I support publication of this comprehensive study if the following points are addressed:

1. In Figure 3A, the differences in the organization of VIM in SRRD WT vs KO condition are not clear. The fragmented appearance that the authors claim is not clearly evident from the images, especially compared to the clearer difference in the distribution of MAP2 and INA in Neuronal cells.

2. In Figure 4A, upon the induction of aggresome formation, the VIM cage is not clearly visible in the images from the rescue experiment and further quantification of the same and hence claimed partial rescue of aggresome formation is unjustified.

3. In Figures 4C to 4F, neither the images nor the line intensity plots clearly show the so-claimed partial rescue of VIM cage and less diffused signal of TDP-43 under SRRD KO+FL SRRD condition.

4. Comparing Figures 6F and S10E, SRRD without NTD still seems to colocalize with TDP-43 and this makes the claim that SRRD acts through its NTDs alone inconclusive

**Have all data underlying the figures and results presented in the manuscript been provided?**

Reviewer #1: Yes

Reviewer #2: **No: **I could not find link or a reviewer token to access the raw sequencing data from their screen.

Reviewer #3: Yes

PLOS authors have the option to publish the peer review history of their article (what does this mean?). If published, this will include your full peer review and any attached files.

Reviewer #1: No

Reviewer #2: No

Reviewer #3: No

---

## [Decision Letter · Decision Letter 1]

15 Jan 2024

Dear Dr Shalem,

We are pleased to inform you that your manuscript entitled "CRISPR screen for protein inclusion formation uncovers a role for SRRD in the regulation of intermediate filament dynamics and aggresome assembly" has been editorially accepted for publication in PLOS Genetics. Congratulations!

The revised manuscript was seen by the three previous reviewers who now recommend acceptance.

Yours sincerely,

Gregory S. Barsh

Editor-in-Chief

PLOS Genetics

Gregory Copenhaver

Editor-in-Chief

PLOS Genetics

Comments from the reviewers (if applicable):

Reviewer's Responses to Questions

**Comments to the Authors:**

Reviewer #1: The authors have done a good job addressing the reviewers concerns and comments. Additional images and cell lines further strengthen the manuscript and edits to the text help to clarify.

Reviewer #2: The authors have addressed all my concerns with additional experimental evidence and text edits.

Reviewer #3: The authors have sufficiently answered all questions and concerns. No further revisions are requested.

**Have all data underlying the figures and results presented in the manuscript been provided?**

Reviewer #1: Yes

Reviewer #2: Yes

Reviewer #3: Yes

PLOS authors have the option to publish the peer review history of their article (what does this mean?). If published, this will include your full peer review and any attached files.

Reviewer #1: No

Reviewer #2: No

Reviewer #3: No

**Data Deposition**

http://datadryad.org/submit?journalID=pgenetics&manu=PGENETICS-D-23-00676R1

**Press Queries**

---

## [Editor Report · Acceptance letter]

29 Jan 2024

PGENETICS-D-23-00676R1 

CRISPR screen for protein inclusion formation uncovers a role for SRRD in the regulation of intermediate filament dynamics and aggresome assembly 

Dear Dr Shalem, 

We are pleased to inform you that your manuscript entitled "CRISPR screen for protein inclusion formation uncovers a role for SRRD in the regulation of intermediate filament dynamics and aggresome assembly" has been formally accepted for publication in PLOS Genetics! Your manuscript is now with our production department and you will be notified of the publication date in due course.

With kind regards,

Judit Kozma

PLOS Genetics

On behalf of:
